



**Radiosounding HARMonization (RHARM): a new homogenized dataset of**
**radiosounding temperature, humidity and wind profiles with uncertainty.**
Fabio Madonna[1], Emanuele Tramutola[1], Souleymane SY[1], Federico Serva[2], Monica Proto[1], Marco Rosoldi[1],
Simone Gagliardi[1], Francesco Amato[1], Fabrizio Marra[1], Alessandro Fassò[3], Tom Gardiner[4], and Peter William
Thorne[5]
[1]Consiglio Nazionale delle Ricerche - Istituto di Metodologie per l'Analisi Ambientale (CNR-IMAA), Tito Scalo (Potenza), Italy
[2]Consiglio Nazionale delle Ricerche – Istituto di Scienze Marine (CNR-ISMAR), Rome, Italy.
[3]University of Bergamo, Bergamo, Italy
[4]National Physical Laboratory, Teddington, UK
[5]Irish Climate Analysis and Research Units, Department of Geography, Maynooth University, Maynooth, Ireland
**Abstract**
Observational records are essential for assessing long-term changes in our climate. However, these
records are more often than not influenced by residual non-climatic factors which must be detected
and adjusted prior to their usage. Ideally, measurement uncertainties should be properly quantified
and validated. In the context of the Copernicus Climate Change Service (C3S), a novel approach,
named RHARM (Radiosounding HARMonization), has been developed to provide a harmonized
dataset of temperature, humidity and wind profiles along with an estimation of the measurement
uncertainties for about 650 radiosounding stations globally. The RHARM method has been applied
to IGRA daily (0000 and 1200 UTC) radiosonde data holdings on 16 standard pressure levels (from
1000 to 10 hPa) from 1978 to present. Relative humidity adjustment and data provision has been
limited to 250 hPa owing to pervasive issues on sensors' performance in the upper troposphere and
lower stratosphere. The applied adjustments are interpolated to all reported significant levels to
retain information content contained within each individual ascent profile. Each historical station
time series is harmonized using two distinct methods. Firstly, the most recent period of the records
when modern radiosonde models have been in operation at each station (typically starting between
2004 and 2010 but varying on a station-by-station basis) are post-processed and adjusted using
reference datasets from the GCOS Reference Upper Air Network (GRUAN) and from the 2010
WMO/CIMO (World Meteorological Organization/Commission for Instruments and Methods of
Observation) radiosonde intercomparison. Subsequently, at each mandatory pressure level, the
remaining historical data are scanned backward in time to detect structural breaks due to prolonged
systematic effects in the measurements and then adjusted to homogenize the time series.
This paper describes the dataset portion related to the adjustment of post-2004 measurements
only. A step-by-step description of the algorithm is reported and comparisons with GRUAN and
atmospheric reanalysis data for temperature and relative humidity data are discussed. The
evaluation shows that the strongest benefit of RHARM compared to existing products is related to
the substantive adjustments applied to relative humidity time series for values below 15% and
above 55% as well as to the provision of the uncertainties for all variables. Uncertainties have been
validated using the ECMWF reanalysis short-range forecast outputs.
The RHARM algorithm is the first to provide homogenized time series of temperature, relative
humidity and wind profiles alongside an estimation of the observational uncertainty for each single
observation at each pressure level. A subset of RHARM data is available at
http://doi.org/10.5281/zenodo.3973353 (Madonna et al., 2020a).



## 1. Introduction

Homogeneous climate data records (CDRs) are essential to diagnosing changes in our climate, understanding its variability, and assessing and contextualizing future climate projections (Cramer et al. 2018). Use of CDRs influenced by residual non-climatic factors may lead to incorrect conclusions about the changing state of the climate (Kivinen et al. 2017). Furthermore, if assimilated within a meteorological reanalysis, these climatic time series may introduce bias instead of positively impacting the final products (Dee et al., 2011). Therefore, when CDRs are used it is important, for any kind of application and to the extent possible, to:

- Detect and adjust for all known and quantifiable systematic inhomogeneities in the observation time series, due to a variety of causes (changes in station location, instrumentation, calibration or drift issues, different instrument sensitivity respect to different networks, changes in the measurement procedures, etc.);
- Establish measurement traceability ideally to an absolute reference (SI or community acknowledged) "standard" through an unbroken chain of calibrations, each contributing to the measurement uncertainty;
- Quantify measurement uncertainties in any data where traceability was not properly established; in such cases, uncertainties must be instead estimated from the available metadata, results of sensors' intercomparisons, or other kinds of information about the measurement process.

Unfortunately, for historical in-situ observations it is often not easy to fulfil the above list of requirements, especially for global baseline or comprehensive networks (Thorne et al., 2017), where the metadata and original pre-processed data (e.g. digital sensor counts) are either missing or retained solely by individual station PIs (if at all) and not routinely shared or stored in their data archives.

This is the case for radiosounding measurements of temperature (T), relative humidity (RH) and wind which still represent anchor information for meteorological reanalysis, despite the advent of GNSS-RO (Global Navigation Satellite System - Radio Occultation) measurements which have proven to be also a very valuable observing system for data assimilation (Bauer et al. 2013). Nevertheless, GNSS-RO measurements are limited in their historical availability, starting only in c.2000. Radiosounding measurements are the only available data source available to study the climate variability in the troposphere and lowermost stratosphere since the mid-20th Century. They also constitute a valuable source of information for satellite cal/val activities (Calbet et al., 2016, Loew et al., 2017, Finazzi et al., 2019). In ERA-Interim ECMWF reanalysis, the conventional observing system which includes radiosoundings, despite proportionately low data volumes, still represents an indispensable constraint (Haimberger et al., 2008). A similar situation exists for the more recent ECMWF ERA5 reanalysis data and for other meteorological reanalysis (Hersbach et al., 2020).

Quality and biases of radiosounding observations strongly varies with sensor type, altitude level, and through time. Several papers have been published using historical radiosounding measurements of temperature to construct CDRs (e.g. Free et al. 2004; Thorne et al., 2005a; McCarthy et al., 2008; Sherwood et al. 2008; Dai et al., 2011; Haimberger et al., 2012). These have used a broad range of approaches enabling an exploration of structural uncertainty (Thorne et al., 2005b). Several products additionally include ensembles which explore parametric uncertainty (Haimberger et al., 2012; Thorne et al., 2011).

A new statistical approach has been recently proposed for future applications (Fassò et al., 2018). Intercomparison datasets made available by various research organizations, institutions and manufacturers represent an invaluable source of information which improves the interpretation of



effects, drifts and any other kind of inhomogeneity in the recorded time series. Most notable of
these are the periodic intercomparison campaigns that have been organized by WMO CIMO which
have typically involved the vast majority of commercial manufacturers (e.g. Nash et al., 2006 or
Nash et al., 2011) providing a thorough snapshot of differences on a periodic basis. These
intercomparisons involve the flying of multiple sonde models on the same rig enabling a direct
comparison of relative performance of the different sensors under the full range of ascent profile
conditions experienced at the location and time of the comparison.
To respond to the need of providing homogeneous and fully traceable upper-air measurements with
quantified uncertainties, the Global Climate Observing System (GCOS) Reference Upper-Air
Network (GRUAN) was established in 2006 (Bodeker et al., 2018). GRUAN aims to provide reference-
quality observations of Essential Climate Variables (ECVs, Bojinski et al., 2014) above Earth's surface.
GRUAN is providing long-term, high-quality radiosounding data at several sites around the world
with characterized uncertainties, ensuring the traceability to SI units or accepted standards,
providing extensive metadata and comprehensive documentation of measurements and
algorithms.
Reference-observing networks provide metrologically traceable observations, with quantified
uncertainty, at small number of stations while baseline-observing networks provide long-term
records that are capable of catching regional, hemispheric and global-scale features, though they
lack absolute traceability (Thorne et al., 2017). As a reference network, GRUAN also provides a basis
for enhanced interpretation of the results, with the quantification of uncertainties, from global
baseline observations. For example, through providing instrumental corrections which can be
extended to non-GRUAN stations to adjust quantifiable systematic effects compromising the quality
of radiosoundings.
The present paper provides an analytic description of the first part of a novel algorithm for
homogenization of historical radiosounding data records available since 1978 (earlier records are
not assessed due to the more heterogeneous data availability at mandatory levels before) which
exploits the added value provided by GRUAN. The approach is named RHARM (Radiosounding
HARMonization) and it is based on two main steps:
1. Adjustment of systematic effects and quantification of uncertainties by post-processing the
radiosounding observations of temperature, humidity and wind since 2004 to present using
the GRUAN data and algorithms as well as the 2010 WMO/CIMO radiosonde
intercomparison dataset [hereinafter ID2010, Nash et al. 2011], made available upon
agreement with WMO;
2. Identification of change-points in the time series and adjustment of non-climatic
(systematic) effects using statistical methods with related quantification of uncertainties in
the historical observations.
The present paper deals with the first part of the RHARM approach which is able to post-process
and adjust a subset of 650 radiosounding stations at the global scale available from the Integrated
Global Radiosonde Archive (IGRA - Durré et al., 2006; Durré et al., 2012). The RHARM dataset
provides a combined homogenization option which is complementary to the limited number of
existing datasets of: homogenized radiosounding temperature measurements, e.g. Radiosonde
Atmospheric Temperature Products for Assessing Climate (RATPAC) by NOAA (Free et al., 2004),
RAdiosonde OBservation COrrection using REanalyses (RAOBCORE), Radiosonde Innovation
Composite Homogenization (RICH) by the University of Wien (Haimberger et al., 2012), Hadley



Centre's radiosonde temperature product v2 (HadAT2) by Met Office (Thorne et al., 2005), Iterative
Universal Kriging v2 (IUKv2) by University of New South Wales (Sherwood and Nishant et al., 2015);
homogenized radiosounding humidity measurements, e.g. the Homogenized RS92 radiosounding
humidity measurements (HomoRS92) by University of Albany (Dai et al., 2011) and the Hadley
Centre's radiosonde temperature and humidity product (HadTH) (McCarthy et al., 2009); and the
only homogenized radiosounding wind dataset "GRASPA" (Ramella-Pralungo et al., 2014a,b).
Distinct from previous efforts, RHARM is the first approach providing the homogenized time series
of temperature, relative humidity and wind in the same package. Moreover, RHARM is based on the
use of "Reference measurements" to calculate and adjust for systematic effects, instead of using
background information provided by meteorological reanalysis, autoregressive models or
neighboring stations. In addition, and of great practical importance, each harmonized data series is
provided with an estimation of the measurement uncertainty. RHARM is also valuable in providing
adjustments on each single radiosounding profile.

The remainder of this paper is organized as follows. In section 2, the data sources used in the paper
are outlined. In section 3, a detailed review of the RHARM data processing for the observations post-
2004 is provided. Specifically, in section 3.1, the algorithms applied for the adjustment of T, RH and
wind profiles measured using Vaisala RS92 radiosondes is outlined, while section 3.2 describes the
adjustments applied to all other radiosonde types than RS92.  In section 4, comparisons between
IGRA, RHARM, GRUAN and ERA5 data are shown and discussed to assess the consistency and
performance of the RHARM algorithm. Section 4.1 compares IGRA, RHARM and GRUAN co-located
data to assess the added-value provided by the RHARM post-processing of IGRA data and to
ascertain the consistency of the RHARM algorithm with the GRUAN Data Processing (GDP). In
section 4.2, comparison between IGRA, RHARM and ERA5 are discussed to quantify inconsistencies
between observational and atmospheric reanalysis data. In section 5, the consistency of the RHARM
estimated uncertainties with the GDP is discussed and a validation of the uncertainties based on the
use of ECMWF forecast model data is presented. Finally, conclusions and an outlook are provided
in Section 6.

## 2. Data sources used
The RHARM approach is applied to the IGRA database which is the most comprehensive collection
of historical and near-real-time radiosonde and pilot balloon observations from around the globe,
maintained and distributed by the National Oceanic and Atmospheric Administration's National
Centers for Environmental Information (NCEI). RHARM is applied to IGRA Version 2 (Durre et al.,
2018) data which was released in 2016 and incorporates data from a considerably greater number
of data sources with an increased data volume by 30% compared to Version 1, extending the data
back in time to as early as 1905, and improving the spatial coverage. IGRA contains observations
from several networks and initiatives, including the GCOS Upper-air Network (GUAN), and the
universal RAwinsonde OBservation program (RAOB). The latter constitutes the largest available
radiosounding data source globally.
From the IGRA data archive, the RHARM approach is applied to a subset of about 650 radiosounding
stations and radiosoundings from ships. The subset consists of those records with documented
metadata (i.e. availability of the radiosonde code, see WMO table 3685, describing the radiosonde
type used at each station over the time) since 2000 (for most of the stations) and for fewer stations
since 1978. For these stations, depending on the used radiosonde type, adjustments based on the
application of GRUAN-like data processing and on the comparison between GRUAN data and ID2010
allow us to provide a quality-enhanced dataset of radiosoundings since 2004, where radiosounding



profiles are corrected for several instrumental effects (e.g. the well-known solar radiation dry-bias).
Beyond the 650 homogenized stations, also the other radiosounding profiles available from IGRA
with documented metadata and a radiosonde model compatible with the GDP or the ID2010 have
been post-processed using RHARM. These additional profiles are provided in the final RHARM
dataset although the paucity of measurements at the considered measurements station does not
allow to complete the homogenization of the corresponding historical time series until 1-1-1978
using RHARM.
The RHARM data harmonization process involves principally the Vaisala RS92 radiosondes (WMO
radiosonde code = 14, 79, 80, 81) launched in the "GRUAN era" (2004-2017) and the Vaisala RS92
NGP (WMO radiosonde code=52), processed in the same way as RS92 on an assumption of
similarity. Since 2016, the Vaisala RS41 sondes are also available (WMO radiosonde code=23, 24,
41, 42) though these are not post-processed by RHARM yet due to the lack, at the present time, of
a specific GRUAN RS41 data product and of any manufacturer independent study on the RS41 data
processing. In Ingleby et al. (2017), operational radiosonde data are compared to ECMWF
background values (12-hour forecast): mean and root-mean-square (rms) Observation-minus-
Background (O-B) statistics show that RS92 NGP sondes have slightly poorer performance
characteristics than the more common RS92 SGP, while there are indications that RS41 perform
slightly better than the RS92. These indications are confirmed by the comparisons shown in Dirksen
et al. (2019) and Madonna et al. (2020b). In Jensen et al. (2018), a comparison is provided between
RS41 and RS92 radiosondes on a limited dataset showing how RS41 does provide important
improvements, particularly in cloudy conditions. GRUAN is currently undertaking a "distributed"
RS41 vs RS92 SGP comparison at its stations, the outcome of which will become available soon
(Dirksen et al., 2020). Following its completion, it is possible that a distinct adjustment approach will
be applied to the RS41 data, if the eventual analysis warrants such a differentiation.
Table 1 gives the number and percentage of radiosonde launches available in the C3S database and
post-processed using the RHARM. Table 1 reveals that more than 85 % of RHARM post-processed
radiosondes are manufactured by Vaisala. On the one hand, this increases the homogeneity of the
dataset globally, whereas on the other hand the dataset is more prone to the impacts of
unquantified random and systematic effects unique to the Vaisala sondes. These can be identified
by comparisons with other datasets such as atmospheric reanalysis (see section 4). The
radiosoundings reported in Table 1 include about 40,000 launches from 37 ships (mostly travelling
in the Atlantic Ocean) processed using RHARM.
There are attendant limitations to the approach proposed above for Vaisala sondes in that: i) the
data processing of Vaisala RS92 radiosoundings provided by IGRA stations is based on the
manufacturer processing software which is used as a black-box and is known to have changed with
time (https://www.vaisala.com/en/sounding-data-continuity). To complicate matters yet further,
the timing when individual stations changed software is not often discernible either from available
(incomplete) metadata or the data series; and ii) raw high resolution profile data from most stations
are not available to allow the full exploitation of the GRUAN data processing methodology despite
their sharing being called for in the latest GCOS Implementation Plan (GCOS, 2016). Furthermore, it
is rarely if ever possible to properly estimate the radiosonde ascent speed from IGRA data due to
missing information of the time of observations (i.e. the observation time at each single pressure
level). Therefore, to apply the radiation correction algorithm proposed in Wang et al. (2013) and
documented in Dirksen et al. (2014), an average ascent speed of 5 m s$^{-1}$ has been assumed. For
these reasons, it is not possible to directly apply the GDP processing as it stands, but only an
approximation, with simplifying assumptions, can be applied.




| Radiosonde type | Launches | Percentage |
|---|---|---|
| LMS6 | 29148 | 1.3 |
| DMF-09 Graw | 16736 | 0.8 |
| VIZ/JinYang | 33721 | 1.5 |
| Taiyuan GTS1-1/GFE(L) | 13409 | 0.6 |
| Nanjing GTS1-2/GFE(L) | 17406 | 0.8 |
| Meteolabor | 436 | 0.0 |
| Meisei | 16179 | 0.7 |
| Beijing Changfeng CF-06 | 36393 | 1.7 |
| M10, Modem | 121446 | 5.5 |
| Vaisala RS92/RS41 | 1893805 | 85.9 |
| Intermet | 26505 | 1.2 |
| Total | 2205183 | 100 |

Table 1: Number and percentage of the radiosonde launches available since 2004 and post-processed using the RHARM
approach. The total number of soundings available within IGRA since 2004 for the stations post-processed using RHARM
is 4,785,543. These include 55,325 balloon launches with a Vaisala RS41 sonde, currently not post-processed within
RHARM.

In Figure 1, the global distribution of the RHARM post-processed stations is shown with the
indication of the 650 station where the homogenization of the historical time series has been
completed. Figure 2, instead, shows the number of post-processed launches at each station.

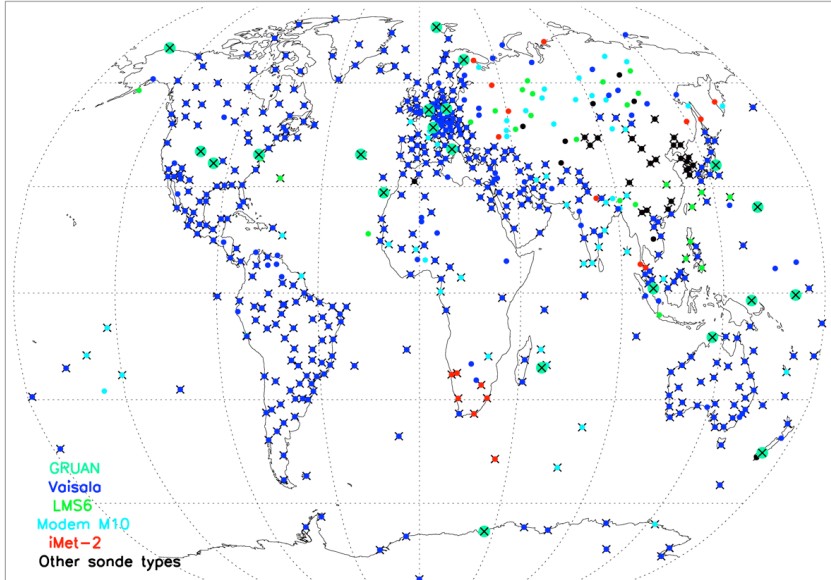

Figure 1: Global distribution of GRUAN Reference stations (green large dots) and the subset of IGRA stations harmonized
using the RHARM approach (small dots). The X symbol indicates the stations where homogenized time series from 1-1-
1978 to present can be obtained using RHARM. The colour legend in the bottom left corner specifies the principal type
of radiosondes used at each station. Some stations have changed sonde types, including switching manufacturers, over
the period of the present analysis.



The global coverage of the RHARM dataset appears reasonably complete, except for Siberia where
a small number of launches is post-processed using RHARM since 2004 on and these prevent the
homogenization of historical time series. The station density in North America, North East Asia, and
East Africa is lower than in Europe, U.S and South America. Nevertheless, the latter regions include
also several stations with the smallest number of available launches, while the stations with largest
number of launches are quite uniformly distributed globally (Figure 2). Table 2 confirms the low
number of measurements available in the Southern Hemisphere (SH), although it is already known
that the quantity of measurements alone cannot address the value of the dataset for a specific study
without a representativeness study (Weatherhead et al., 2017).

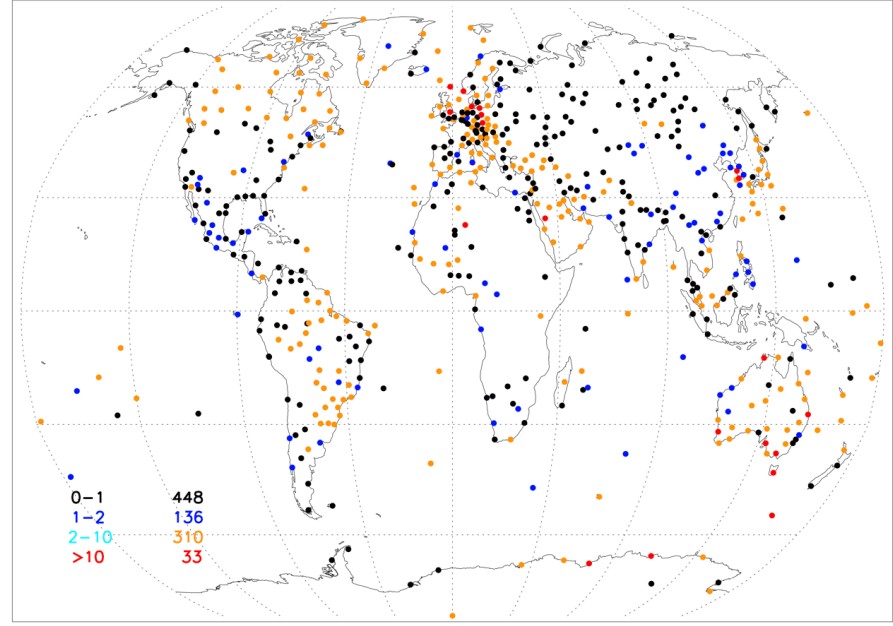

Figure 2: Quantity of RHARM post-processed radiosoundings available. The scale in the left bottom corner denotes
available radiosoundings at each station (in thousands of ascents).

| Region | Latitude range | Number of adjusted launches (thousands) | Percentage |
|---|---|---|---|
| Arctic | 70N-90N | 70.3 | 3.2 |
| Northern Hemisphere mid-latitudes | 25N-70N | 1177.9 | 53.4 |
| Tropics | 25N-25S | 611.3 | 27.7 |
| Southern Hemisphere mid-latitudes | 25S-70S | 325.7 | 14.8 |
| Antarctic | 70S-90S | 20.0 | 0.9 |
| **Total** | | **2205.2** | **100** |

Table 2: Number of launches at different latitudes for the stations shown in Figure 1. In the last column of Table 2, the
percentage with respect to the total number available globally is also provided.



Further in-depth statistical analysis of the IGRA Version 2 historical times series and their temporal
and spatial coverage at different pressure levels is available in Durre et al. (2018) and in Ferreira et
al. (2019), the latter for relative humidity observations only. Another statistical analysis of the
missing data and of their spatial coverage is provided in Sy et al. (2020).
Many of the RHARM stations are GCOS Upper Air Network (GUAN) sites with a commitment to long-
term operation, a guideline that at least 25 radiosonde launches per month should reach 30 hPa,
and an articulated aim for compliance with best practice for GUAN stations, although in reality they
very frequently fall short of these requirements (for more information see
http://www.wmo.int/pages/prog/gcos/index.php?name=ObservingSystemsandData). Within
ECMWF or other NWP systems (Dee et al., 2011; Ingleby et al., 2017), GUAN stations are not
distinguished from the broader RAOB network, though ECMWF does monitor data availability from
GUAN separately on behalf of GCOS (http://www.ecmwf.int/en/forecasts/quality-our-
forecasts/monitoring-observing-system). Some GUAN stations have not reported observations for
long periods, while for others there may be temporary outages. In some cases, GUAN stations
provide radiosounding profiles to greater heights than neighboring ones or two ascents a day rather
than one. Recent work has assessed the quality of different radiosonde types by examining O-B
departures for the two-year period 2015-2016 (Ingleby et al., 2017). GUAN and RAOB data has
shown very similar performance. As the two networks are not sufficiently distinguishable, they are
considered fully equivalent for the purposes of RHARM. There is presently an ongoing GCOS task
team for GUAN (https://library.wmo.int/doc_num.php?explnum_id=4469), which may eventually
provide a basis to distinguish between future GUAN operations and the broader RAOB program.
Furthermore, the nascent Global Basic Observing Network if adopted and fully implemented may
provide another reason to differentiate between stations based upon network affiliation in future.
Future updates of RHARM could reconsider the decision to treat as equivalent all such
measurements should developments require such a re-evaluation.

## 3. Methodology
The RHARM homogenization of global radiosounding temperature, humidity and wind profiles is
applied to daily (00:00 and 12:00 UTC) radiosonde data on 16 mandatory pressure levels (10, 20, 30,
50, 70, 100, 150, 200, 250, 300, 400, 500, 700, 850, 925, 1000 hPa) arising from the IGRA database.
Relative humidity (RH) adjustments are limited to 250 hPa owing to pervasive sensor performance
issues at greater altitudes. Profiles are post-processed at these mandatory pressure levels which do
not change on a per-profile basis, as occurs for significant levels. The applied adjustments are then
interpolated to the significant levels. Uncertainties are estimated for each processing step listed
above and propagated to estimate the total uncertainty.

It is important to note that adjustments at 1000 hPa and all levels above 10 hPa (<10hPa), due to
the small sizes of the available observation sample, must be handled with care because they are less
representative of the real differences at the corresponding altitudes.

### 3.1 Adjustment of Vaisala temperature, humidity and wind profiles
During daytime, the sensor boom of any radiosonde type is heated by solar radiation which
introduces biases in temperature and humidity (Wang et al., 2013). The net heating of the
temperature sensor and the dry-bias affecting the relative humidity sensors depends on the amount



of absorbed radiation and, therefore, the solar elevation angle (α), as well as on the cooling by
thermal emission and ventilation by air flowing around the sensor (Dirksen et al., 2014).
To adjust this effect in the measured profiles of temperature and RH, the first step of the RHARM
algorithm, involving only the Vaisala RS92 sondes, is to apply a solar radiation correction to the T
vertical profiles (all levels, mandatory and significant) in a way similar to the GDP. This is performed
in two steps:
1.  first, the radiation correction, $\Delta T_{VAISALA}$, applied (subtracted) by the manufacturer (Vaisala)
323         to the temperature profiles is removed;
2.  second, a GRUAN-like radiation correction, $\Delta T_{GRUAN}$, is calculated using the values of the
325         actinic flux modelled with the Streamer RTM (Key and Schweiger, 1998) and applied to the
326         RS92 sondes. Where GRUAN-like corrections cannot be applied, the manufacturer
327         correction is left unchanged.

$\Delta T_{VAISALA}$ is derived from the tables provided by the manufacturer and accounts for the changes to
the RS92 data processing during the sonde model's production lifetime (see
https://www.vaisala.com/en/sounding-data-continuity).
The GRUAN correction, $\Delta T_{GRUAN}$, is defined as:
$$\Delta T_{GRUAN}(I_a, p, v) = ax^b \quad [Eq.1]$$
$$x = \frac{I_a}{pu} \quad [Eq.2]$$
where $I_a$ is the actinic flux at the solar zenith angle of the balloon release time, calculated using the
LOWTRAN          v7         solar          position         data         (taken         from
https://code.arm.gov/vap/mfrsrod1barnmich/blob/ed71a3666e8e1781ed8d753e859b284f3b7dcc
2e/src/zensun.pro); p is the pressure level; and u is the ascent speed in m s$^{-1}$. Unfortunately, u
cannot be directly ascertained from IGRA data due to the missing reporting of the observation time
at each pressure level for most soundings. For this reason, an average value of 5 m s$^{-1}$ for the ascent
speed is assumed in the RHARM approach. This corresponds to the recommended ascent speed
from WMO guidance and corresponds well to known profile ascent speeds (e.g. Madonna et al.,
2020b). The coefficients a and b in Eq.1 are fit parameters arising from laboratory experiments
(Dirksen et al., 2014) yielding a = 0.18(±0.03) and b = 0.55(±0.06).
Once $\Delta T_{GRUAN}$ is calculated, the final correction applied by GRUAN to the T profiles following the
approach in Dirksen et al. (2014) is to derive a best estimate that lies between the two approaches:
$$\Delta T = \frac{(\Delta T_{GRUAN} + \Delta T_{VAISALA})}{2} \quad [Eq.3]$$
Within RHARM, the final adjustment added to IGRA temperature profiles is:
$$\Delta T_{RHARM,RS92} = \Delta T_{VAISALA} - \Delta T + \Delta T_r \quad [Eq.4]$$
where $\Delta T_r$ is a residual calibration bias calculated from the mean difference of GRUAN and IGRA
night time temperature profiles at mandatory pressure levels for the six GRUAN sites reported in



Table 3. To calculate $\Delta T_r$, outliers are filtered using a robust Z-score method. $\Delta T_r$ is added to both
night and daytime profiles. If the value of Ia in equation 2 is equal to zero (i.e. $\Delta T$=0), the
manufacturer radiation correction applied to IGRA profiles is not modified and Eq.4 reduces to
$\Delta T_{RHARM,RS92} = \Delta T_r$. Eq. 4 allows to remove the solar radiation correction applied by the
manufacturer and to adjust the data using the GRUAN correction plus an additional correction
whose aim is to reduce, on average, the gap with the GDP.
The uncertainties on $T_{RHARM,RS92}$, $\varepsilon(T_{RHARM,RS92})$, are calculated according to the following
equation:
$\varepsilon(T_{RHARM,RS92}) = \left(\varepsilon_{c,I_a}(\Delta T)^2 + \varepsilon_{c,R_c}(\Delta T)^2 + \varepsilon_{vent}(\Delta T)^2 + \varepsilon_r(\Delta T)^2 + \varepsilon_R(\Delta T)^2\right)^{\frac{1}{2}}$          [Eq.5]
In Eq. 5, $\varepsilon_{c,I_a}(\Delta T)$ is the uncertainty due to the estimation of the solar actinic flux; $\varepsilon_{c,R_c}(\Delta T)$ is the
uncertainty due to parameters estimated in the radiation correction model reported in Eq. 1.
Formulas to calculate $\varepsilon_{c,I_a}(\Delta T)$ $and$ $\varepsilon_{c,R_c}(\Delta T)$ are fully documented in Dirksen et al. (2014). $\varepsilon_{vent}$ is
the uncertainty due to the ventilation rate (including the effect of the pendulum motion of the
radiosonde assumed as in GRUAN to be of about 0.2 m s$^{-1}$); $\varepsilon_r$ is used to indicate the comparison
uncertainties estimated from the standard deviation of $\Delta T_r$; $\varepsilon_R$ is an additional random uncertainty
added to the profiles of 0.15 K in agreement with the GDP approach (Dirksen et al., 2014), although
for RHARM this cannot be quantified as done by GRUAN due to the unavailability of raw data. When
the radiation correction of the manufacturer is left unchanged, $\varepsilon(T_{RHARM,RS92})$ is assumed to be the
same as the closest temperature profile in time measured under the same meteorological
conditions (i.e. clear sky or cloudy).

| GRUAN code | Station name and country | Latitude | Longitude | Altitude | WMO index |
|---|---|---|---|---|---|
| CAB | Cabauw, Netherlands | 51.97° | 4.92° | 1 m | 06260 |
| LIN | Lindenberg, Germany | 52.21° | 14.12° | 98 m | 10393 |
| NYA | Ny-Ålesund, Norway | 78.92° | 11.92° | 5 m | 01004 |
| SGP | Lamont, OK, USA | 36.60° | -97.49° | 320 m | 74646 |
| SOD | Sodankylä, Finland | 67.37° | 26.63° | 179 m | 02836 |
| TAT | Tateno, Japan | 36.06° | 140.13° | 25 m | 47646 |

Table 3: List of the GRUAN stations used to calculate the additional calibration bias applied in the RHARM approach to
adjust the Vaisala RS92 radiosoundings available from IGRA.

Following the application of temperature adjustments, the measured value of the relative humidity
(all levels), $RH_{RHARM,RS92}$ is adjusted for the solar radiation dry-bias, estimated by the effect of the
T warm bias on the saturation vapor pressure, using a correction factor calculated using the
following formula:

$RH_{RHARM,RS92} = cf\, RH_{IGRA,RS92} \left(\frac{p_s(T_{RHARM,RS92} + f\Delta T_{RHARM,RS92})}{p_s(T_{RHARM,RS92})}\right)$ [Eq.6]

where $cf$ is scalar a factor accounting for the temperature dependency of the sensor calibration
estimated at night by a comparison with GRUAN measurements; $p_s$ is the saturation vapor pressure
and $f$ is a factor determined experimentally to weight the applied correction on different



radiosonde batches (Dirksen et., 2014). The factor *cf* may embed a residual contribution from the
sensors' time-lag which is typically small for the RH values up to 250 hPa. For the sake of clarity, a
flow diagram describing the application of the RHARM adjustments to both T and RH profiles is
shown in Figure 3.

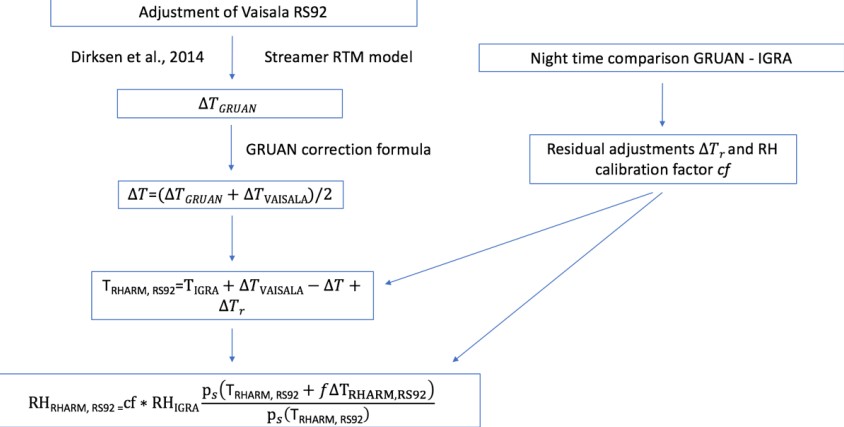

Figure 3: Flow diagram summarizing the post-processing steps of the RHARM algorithm to adjust temperature and
relative humidity profiles measured by the RS92 sondes since 2004. In the diagram, cf is a calibration factor, $p_s$ is the
saturation vapor pressure, f is a factor determined experimentally to weight the applied correction on different
radiosonde batches used over the years. ΔT indicates the adjustments applied to temperature, ΔRH to relative humidity.
The subscripts refer to the GRUAN adjustments, IGRA adjustments (manufacturer-based plus IGRA quality control),
RHARM adjustments and to RS92 Vaisala sondes. The subscript "r" refers to a residual correction derived from the night
time comparison between GRUAN and IGRA data at six GRUAN sites, reported in Table 3.

The uncertainties on $RH_{RHARM,RS92}$, $\varepsilon(RH_{RHARM,RS92})$, are calculated according to the following
equation:
$$\varepsilon(RH_{RHARM,RS92}) = \left(\varepsilon_{RC_T}(\Delta RH)^2 + \varepsilon_{RC_f}(\Delta RH)^2 + \varepsilon_{cf}(\Delta RH)^2 + \varepsilon_R(\Delta RH)^2\right)^{\frac{1}{2}} \text{ [Eq.7]}$$
In Eq. 7, $\varepsilon_{RC_T}(\Delta RH)$ is the uncertainty of dry bias correction and $\varepsilon_{RC_f}(\Delta RH)$ is the uncertainty of
the radiation sensitivity factor f in Eq. 5; $\varepsilon_{cf}$ is the uncertainty due to calibration factor cf; $\varepsilon_R$ is an
additional random uncertainty of 2% RH. In analogy with temperature, when the radiation
correction of the manufacturer is left unchanged, $\varepsilon(RH_{RHARM,RS92})$ is assumed to be the same as
the closest RH profile in time measured under the same meteorological conditions.
At present, there are only two GRUAN data products (GDP), for the Vaisala RS92 and for Meisei
RG11 sondes. RHARM applies adjustments to RS92 Vaisala sondes only, which represents a
substantive portion of the global data. For the Meisei RG11 GDP, its recent introduction (Kobayashi
et al., 2019) has not allowed yet the implementation within RHARM, but an update of the data
processing will be implemented in the near future along with any other GRUAN GDP which might
become available.
It is important to note that at the end of 2010, Vaisala operational processing underwent a major
change with the inclusion of humidity time-lag correction and an improved dry-bias correction for
RH. Stations applied this update to the software in a heterogeneous way. For example, Germany
and the UK started using it in 2015, some others earlier and others later or not at all, due to different
choices by the NMSs. In this version of RHARM it is very difficult to take into account such changes



at each individual station given the grossly insufficient metadata available. Nevertheless, this may
be possible in future, for any such subsequent changes, using native BUFR reports which have the
version number of the processing in their extra metadata. Storing of these files on a routine basis
has been undertaken by ECMWF starting from 2016. An effort to cooperate with Vaisala will also be
undertaken to identify when individual stations switched, in order to improve future updates of the
RHARM dataset.
Differently from temperature and relative humidity data, the GDP on wind profiles is more basic
and does not apply as many corrections to the raw data. The manufacturer software retrieves the
magnitudes of u and v from the Doppler shift in the GNSS carrier signal. In the GRUAN processing,
these vectors are smoothed and converted into wind speed and direction. The noise in the raw data
of u and v, due to the radiosonde's pendulum motion and the noise of the GNSS data, is reduced by
using a low-pass digital filter (Dirksen et al., 2014). This smoothing reduces the effective temporal
resolution of the wind data to 40 s. Using statistical uncertainties calculated for u and v, the
uncertainty of the wind direction $\phi$ is given by:

$$\varepsilon(\phi) = \frac{180}{\pi} \frac{\sqrt{\delta_u^2 + \delta_v^2}}{\left(1 + \left(\frac{u}{v}\right)^2\right)|v|} \quad \text{[Eq. 8]}$$

and the uncertainty of the wind speed $w$ by

$$\varepsilon(w) = \sqrt{\frac{(u\delta_u)^2 + (v\delta_v)^2}{u^2 + v^2}} \quad \text{[Eq. 9]}$$


Typical values are between 0.4 and 1 ms[-1] for $\varepsilon(w)$ and about 1° for $\varepsilon(\phi)$. In the case of negligible
wind, when u and v approach 0, the value of $\varepsilon(\phi)$ becomes very large. For such cases, the absolute
value of $\varepsilon(\phi)$ is limited to 180° (Dirksen et al., 2014). The same limitation is applied to uncertainties
estimated with RHARM.
The RHARM algorithm converts wind direction and speed reported in IGRA data files into the
vectorial components u and v. At time instant t and at a pressure level p, these variables are related
as follows:

$$u(p, t) = w(p, t) \sin\left(\frac{\pi}{180}\phi(p, t)\right) \quad \text{[Eq. 10]}$$


$$v(p, t) = w(p, t) \cos\left(\frac{\pi}{180}\phi(p, t)\right) \quad \text{[Eq. 11]}$$


The conversion into u and v components avoids issues of interpretation over averages or differences
associated with the use of the discontinuous wind direction scale. Nevertheless, to facilitate use
applications preferring the use of wind speed and direction, a final step of the processing converts
the vectors back into wind speed and direction. Eqs. 8 and 9 are then used also in RHARM to
estimate the final uncertainty on w and $\phi$.

To adjust the IGRA wind profiles, the day and night time differences for u and v between the GRUAN
processed and the IGRA radiosounding wind profiles have been calculated using the stations in Table





1. The approach is the same as for temperature and relative humidity, although Eq. 4 is reduced to
$\Delta u_{RHARM,RS92} = \Delta u_r$ and to $\Delta v_{RHARM,RS92} = \Delta v_r$, for each of the wind vectorial components. The
standard deviation of the $\Delta u_{RHARM,RS92}$ and $\Delta v_{RHARM,RS92}$ are then used as the estimation of the
adjustment uncertainties, which will be expressed as $\varepsilon\left(\Delta u_{RHARM,RS92}\right) = \left(\varepsilon_r(\Delta u)^2 + \varepsilon_R(\Delta u)^2\right)^{\frac{1}{2}}$
and $\varepsilon\left(\Delta v_{RHARM,RS92}\right) = \left(\varepsilon_r(\Delta v)^2 + \varepsilon_R(\Delta v)^2\right)^{\frac{1}{2}}$. $\varepsilon_R$ is a random uncertainty of 01.5 m s$^{-1}$ for both u
and v.
This adjustment can only partly reconcile the difference between GDP and manufacturer data
processing at all the sites because typically the difference is higher, due to the differences in the
low-pass filtering applied to reduce the effect of the radiosonde's pendulum motion.
The adjustment applied to temperature, humidity and wind profiles at the mandatory levels as well
as the corresponding uncertainties are finally interpolated at the significant levels available in the
IGRA files, which varies from profile-to-profile and is used to mark significant geophysical points in
the profile such as temperature or humidity profile inflections. The interpolation is performed using
a linear function for temperature, while a cubic spline interpolation has been applied to RH and
wind component profiles. The resulting interpolation uncertainty has been evaluated using the
comparison of the effect of the interpolation at GRUAN stations where high resolution profiles are
available. This interpolation uncertainty has been added to the final uncertainty budget (for T,
$\sigma$=0.25 K, for RH, $\sigma$=0.5 %, for both u and v, $\sigma$=0.05 ms$^{-1}$).

## 3.2 Adjustment of other radiosonde types
Section 3.1 described the adjustments applied to the RS92 sondes, which represent the main link of
RHARM to GRUAN data and the GDP. For remaining radiosonde types, the adjustment estimation
requires the adoption of a different approach due to the unavailability of GRUAN reference products
for the vast majority of radiosonde types other than Vaisala RS92. To harmonize these records,
RHARM makes primary recourse to the ID2010, which is a unique dataset from which estimations
of the performance of operational radiosondes in 2010 were evaluated through a joint effort
between the scientific community and the various manufacturers. ID2010 allows us to assess the
systematic component of the inter-sensor differences, it does not contain strong outliers, but the
post-processing applied may come at the cost of under-representing sonde-to-sonde random
uncertainty effects (Nash et al., 2011). Furthermore, the use of complex multi-sonde rigs may alter
the sonde characteristics compared to standard single-payload flights in important ways vis-a-vis
aspects such as ventilation, thermal effects and the magnitude and periodicity of any pendulum
effects.
Among the radiosonde types involved in the intercomparison, only those routinely employed at a
sufficient number of stations worldwide have been considered for calculating the adjustments for
RHARM. The Vaisala RS92-SGP (WMO radiosonde code=80) was used as one of the common models
during (almost) all flights, allowing us to tie each sonde to the RS92 (at least for the particular
location, RS92 model version, the most recent update in the RS92 Vaisala data processing in
operation at the time, and the season of the campaign). In addition to keeping consistency with one
of only two reference products currently available through GRUAN, Vaisala RS92 sondes available
in ID2010 have been post-processed using the RHARM algorithm. The list of the selected radiosonde
types is given in Table 4.



Due to the launching setup adopted during the WMO intercomparison, a few radiosonde types were
compared less frequently than others on the same payload. Specifically, there was a subset of
models which did not have a sufficient sample of Vaisala RS92 sondes associated. In these cases,
the Graw radiosondes, which flew on sufficient rigs both with RS92 sondes and the under-sampled
sondes, have been used to make the bridge with the RS92 and to calculate statistics on a larger
number of comparisons. Standard deviations have been recalculated accordingly to consider the
additional contribution of the Graw radiosonde uncertainties and the two-steps required. The mean
difference between RS92 temperature profiles and the profiles measured by each of the sondes
listed in Table 4 (hereinafter named as "NORS92") has been quantified as:

$$\Delta T_{NORS92} = \frac{1}{N} \sum_{i=1}^{N} T_i^{NORS92} - T_i^{RHARM,\,RS92}\ \text{[Eq. 12]},$$

and the standard deviation $\sigma_{T_{NORS92}}$ is calculated from the spread of pairwise estimates of $\Delta T_{NORS92}$
arising from the RHS term of equation 12. $\sigma_{T_{NORS92}} = \sqrt{\sigma_{T_{NORS92}}{}^2 + \varepsilon\left(T_{RHARM,RS92}\right)^2}$ is used as the
best estimate of the uncertainty for $\Delta T_{NORS92}$ . If the Graw radiosonde is considered as the link with
the Vaisala RS92, Eq.12 becomes:

$$\Delta T_{NORS92} = \left(\frac{1}{N} \sum_{i=1}^{N} T_i^{NORS92} - T_i^{GRAW}\right) - \left(\frac{1}{M} \sum_{j=1}^{M} T_j^{GRAW} - T_j^{RHARM,\,RS92}\right)\ \text{[Eq. 13]},$$

Although the ID2010 have already been processed for the presence of outliers, $\Delta T_{NORS92}$ and
$\sigma_{T_{NORS92}}$ have been calculated using a resistant algorithm where the mean trims away outliers using
the median and the median absolute deviation
(https://idlastro.gsfc.nasa.gov/ftp/pro/robust/resistant_mean.pro). This allows us to ensure that
the most typical differences between two radiosonde types are caught in the calculated differences,
enabling their application as an average adjustment on a wide range of radiosondes. Eqs. 12 and 13,
with the related considerations, are applied also to wind profiles.

| Abbrev. | Name | WMO radiosonde code |
|---------|------|---------------------|
| RS92 | VAISALA RS92 SGP | 80 |
| Graw | DMF-09 Graw | 17 |
| Modem | M10, Modem | 57 |
| LM | LMS6 | 11 (01/01/2008), 82 (07/11/2012) |
| Meisei | Meisei | 30 (01/01/2010) |
| JinYang | JinYang | 21 |
| IntermSA | iMet-2 InterMet | 97, 98, 99 |
| Daqiao | Nanjing GTS1-2/GFE(L) | 33 (03/11/2011) |
| Huayun | Taiyuan GTS1-1/GFE(L) | 31 (03/11/2011) |
| Changf | Beijing Changfeng CF-06 | 45 (07/05/2014) |
| ML | Meteolabor | 26 |

Table 4: List of the operational radiosondes involved in the 2010 WMO/CIMO radiosonde intercomparison which have
been used to calculate the RHARM adjustments. Dates in brackets are referred to the date of assignment for the WMO
radiosonde code. Please note that also RS92 is included in the list. Adjustments have been calculated using the RS92-
SGP sondes as the reference, in order to be physically consistent with the GRUAN product. For consistency, RS92-SGP
sondes launched during the intercomparison have been reprocessed using the RHARM post-processing approach.




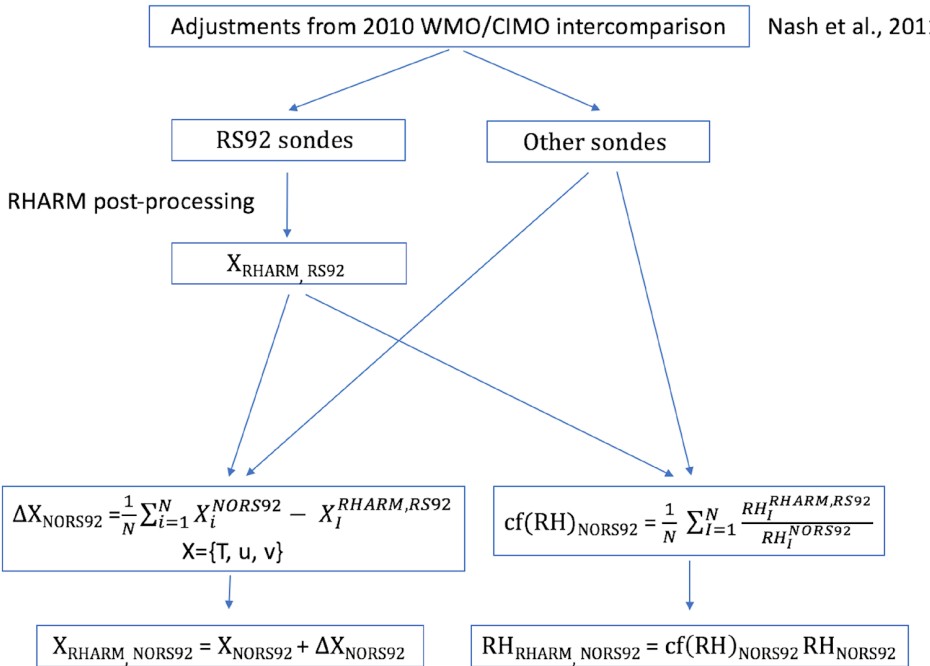

Figure 4: Flow diagram summarizing the post-processing steps of the RHARM algorithm to the adjust the temperature
and relative humidity profiles measured for all radiosonde types other than RS92 reported in Table 4 in the period since
2004 onward. In the diagram, "X" stands for T, u or v. The subscript RHARM refers to the output adjusted variable and
the subscripts RS92/NORS92 refer to the input radiosonde type: RS92 Vaisala or other.

For relative humidity, also in order to be consistent with the RHARM post-processing of RS92
sondes, instead of Eq. 12 the following is used:


$$cf(RH)_{NORS92} = \frac{1}{N} \sum_{i=1}^{N} \frac{RH_i^{RHARM,RS92}}{RH_i^{NORS92}} \text{ [Eq. 14]},$$

where $cf(RH)_{NORS92}$ is a scalar calibration factor to remove systematic effects on the NORS92
radiosondes; the related standard deviation, $\sigma_{cf(RH)_{NORS92}}$, is calculated via error propagation. If the
Graw radiosonde is considered as the link with the Vaisala RS92, Eq.14 becomes:

$$cf(RH)_{NORS92} = \frac{1}{N} \sum_{i=1}^{N} \frac{cf(RH)_{GRAW} \; RH_i^{RHARM,GRAW}}{RH_i^{NORS92}} \text{ [Eq. 15]},$$

To facilitate the application of the adjustments for all significant pressure levels available in the IGRA
dataset, the profiles obtained from the Eqs. 12, 13, 14 and 15, including all the available (mandatory
and significant) levels, have been first smoothed to an effective resolution of 100 m (Iarlori et al.,
2015), to reduce the uncertainties due to the limited sample size, and then interpolated at 0.1 hPa
resolution. Interpolation has been performed to allow the processing chain to always get an exact
match with any of the mandatory and significant levels available in the IGRA files. As for the
significant levels reported in the RS92 radiosonde profiles, the interpolation has been performed
using a linear function for temperature, while a cubic spline interpolation has been applied to RH
and wind component profiles. The interpolation uncertainty has been finally added to the final
uncertainty budget (for T, $\sigma$=0.25 K, for RH, $\sigma$=0.5 %, for both u and v, $\sigma$=0.05 ms$^{-1}$).

In Figure 5, $\Delta T_{RS92,NORS92}$ is shown with the corresponding standard deviations $\sigma_{\Delta T_{RS92,NORS92}}$ for
ten radiosonde types during night (upper panels) and day (lower panels) up to 50 hPa. $\Delta T_{RS92,NORS92}$
ranges between -0.2 K and 0.3 K up to 200 hPa, both at night and day. At higher altitudes,
$\Delta T_{RS92,NORS92}$ increases with values between -0.3 K and 0.6 K. For a few radiosonde types, the
ID2010 provides only a few profiles to calculate the adjustments up to 50 hPa and beyond. This may
strongly increase the value of $\Delta T_{RS92,NORS92}$ and of the related standard deviation. For this reason,
the profiles in Figure 5 have been cut at tailored pressure levels $p_t$ (ranging between 30 hPa and 100
hPa) and at pressures lower than $p_t$ the adjustment applied in RHARM is equal to the value of
$\Delta T_{RS92,NORS92}$ at $p_t$. $\sigma_{\Delta T_{RS92,NORS92}}$ is within 0.2 K at night up to 200 hPa and increases to 0.3-0.4 K at
100 hPa. A couple of radiosonde types show a larger standard deviation (e.g. JinYang). During
daytime $\sigma_{\Delta T_{RS92,NORS92}}$ is larger than at night but is still less than 0.3K up to 200 hPa, while values
above this level are very similar to nighttime. The Meisei comparison profiles appear to be generally
noisier than the other types, particularly during the day. Is it also worth noting that some of the
apparent periodicity in the left panels of Figure 5 are likely relate to manufacturer-to-manufacturer
differences in accounting for the effect of the pendulum motion of the radiosondes.

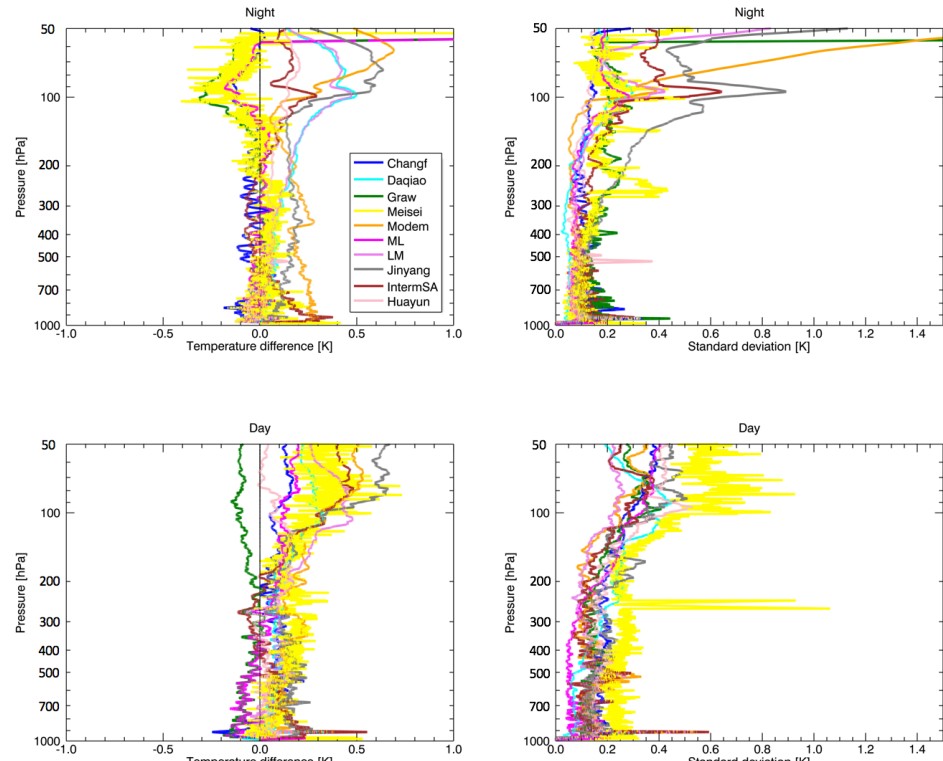

Figure 5: Left panels, night time and daytime profiles of the mean differences between RS92 temperature profiles and
the profiles measured by all the other radiosonde types listed in Table 4; right panels, profiles of the standard deviation
of the mean difference, reported in the corresponding left panels.

In Figure 6, the mean difference $\Delta RH_{NORS92} = \frac{1}{N}\sum_{i=1}^{N} RH_i^{RHARM, RS92} - RH_i^{NORS92}$ is shown with
the corresponding standard deviation. The values of $\Delta RH_{NORS92}$ are shown instead of
$cf(RH)_{NORS92}$, which is the factor calculated in Eq.15, to give a clearer quantitative representation
of the difference among the various radiosonde types for the ID2010. The plots in Figure 6 are shown
up to 250 hPa which is the maximum altitude at which the RHARM approach performs the post-
processing. $\Delta RH_{NORS92}$ ranges within about ±10% from the surface up to 500 hPa, both at night and
day, although it is mostly positive for all radiosonde types during the day: this indicates that the
adjustments applied to correct the effect of solar radiation by most of the manufacturers
underestimates the RH profiles compared to the RHARM processed Vaisala RS92 profiles. At
pressure levels above 500 hPa, $\Delta RH_{NORS92}$ generally increases with altitude and is positive during
the day. The only exception is the Modem radiosondes which at night exhibit negative values of
$\Delta RH_{NORS92}$, smaller than -15%, and Daqiao and Meteolabor for very few levels at pressures higher
than 300 hPa. $\sigma_{\Delta RH_{NORS92}}$ is smaller than 10% at night and day, except for a few larger values at
levels below 400 hPa reported for the Daqiao, Huayun and Meteolabor radiosondes.

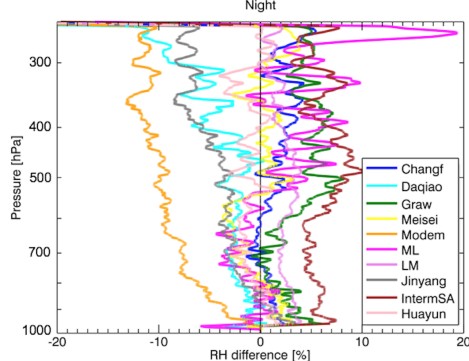
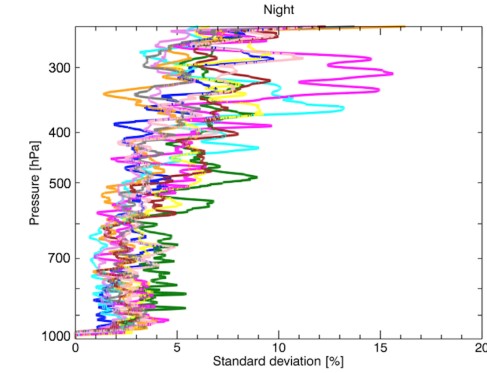
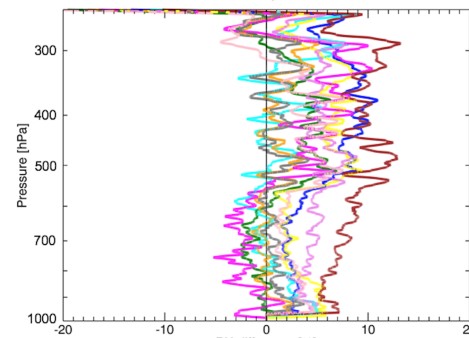
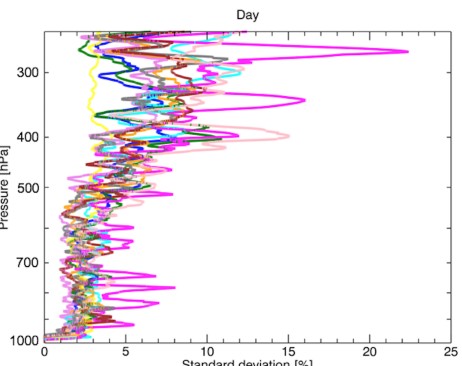

Figure 6: Same as Figure 5 but for RH.

In analogy with Figures 5 and 6, Figure 7 shows the profiles of $\Delta u_{RS92,NORS92}$ with the corresponding
standard deviations $\sigma_{\Delta u_{RS92,NORS92}}$. The ID2010, apart from the Daqiao sondes, includes only winds
measurements based on GNSS tracking of the radiosonde. Moreover, in the ID2010 daytime and
night time measurements were treated together as no significant difference could be found



between the two categories. Nevertheless, considering that a different approach to the processing
of the ID2010 is adopted by RHARM (i.e. decomposition into vectorial wind components u and v)
and that here only one radiosonde model (e.g. RS92) is assumed as the reference for the calculation
of adjustment profiles for all other sonde types of the ID2010, we treated daytime and night time
data separately in order to check the robustness of the estimated adjustments.
At night, $\Delta u_{RS92,NORS92}$ is predominantly negative throughout the profile for all manufacturers, but
smaller than -0.5 ms$^{-1}$ up to 400 hPa, then increases up to -2.0 ms$^{-1}$ at 100 hPa reaching its maximum
value. During the day, the same behavior is observed although the values from the surface to 400
hPa show greater spread. $\sigma_{\Delta u_{RS92,NORS92}}$ is lower than 2.0 ms$^{-1}$ for both day and night, except for
Graw and Modem radiosondes above 100 hPa and 50 hPa heights, respectively. Figure 8 shows the
same as Figure 7 but for $\Delta v_{RS92,NORS92}$. Both at night and day, $\Delta v_{RS92,NORS92}$ is negative and smaller
than -0.5 ms$^{-1}$ up to 400 hPa while it is positive at lower pressure levels with values lower than 1.0
ms$^{-1}$. The small sample size for the comparison clearly affects the values of $\Delta v_{RS92,NORS92}$ at levels
above 100 hPa. The same is true for $\sigma_{\Delta v_{RS92,NORS92}}$ for Graw and Modem sondes at night.
$\sigma_{\Delta u_{RS92,NORS92}}$ is generally lower than 1.0 ms$^{-1}$ both at night and day.

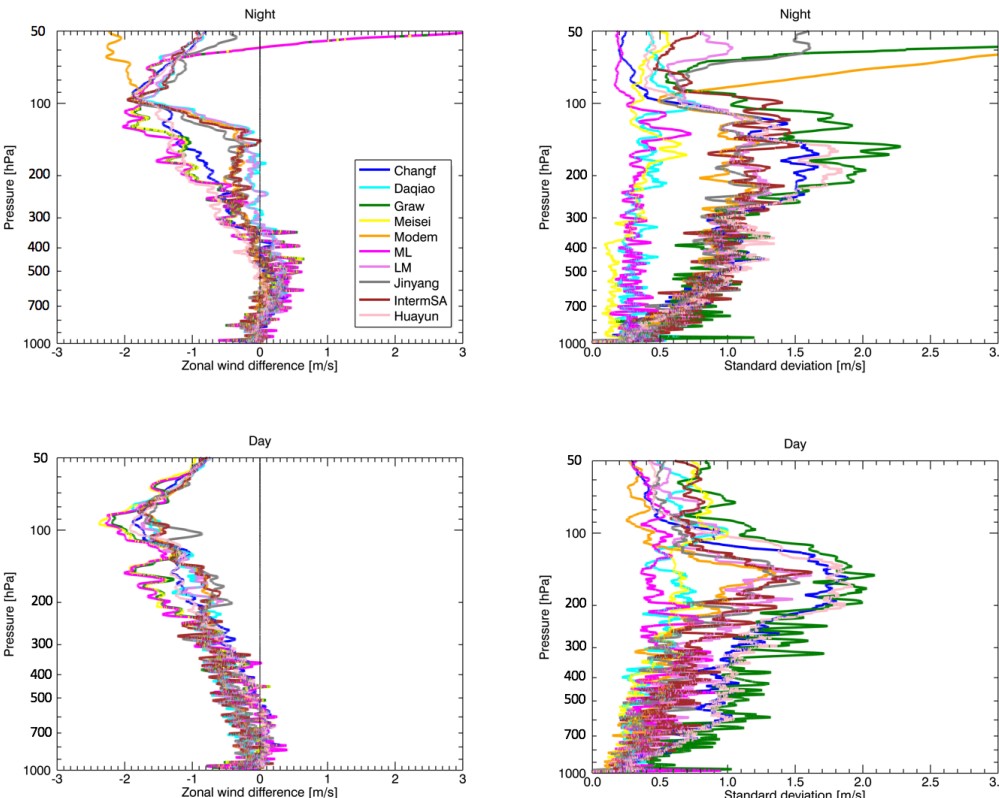


Figure 7: Same as Figure 5 but for the zonal wind component (u).

As for temperature, $\Delta u_{RS92,NORS92}$ and $\Delta v_{RS92,NORS92}$ profiles have been cut at tailored pressure
levels $p_t$ and at pressure levels lower than $p_t$ the adjustment is equal to the value of $\Delta u_{RS92,NORS92}$
and $\Delta v_{RS92,NORS92}$ at $p_t$, respectively.

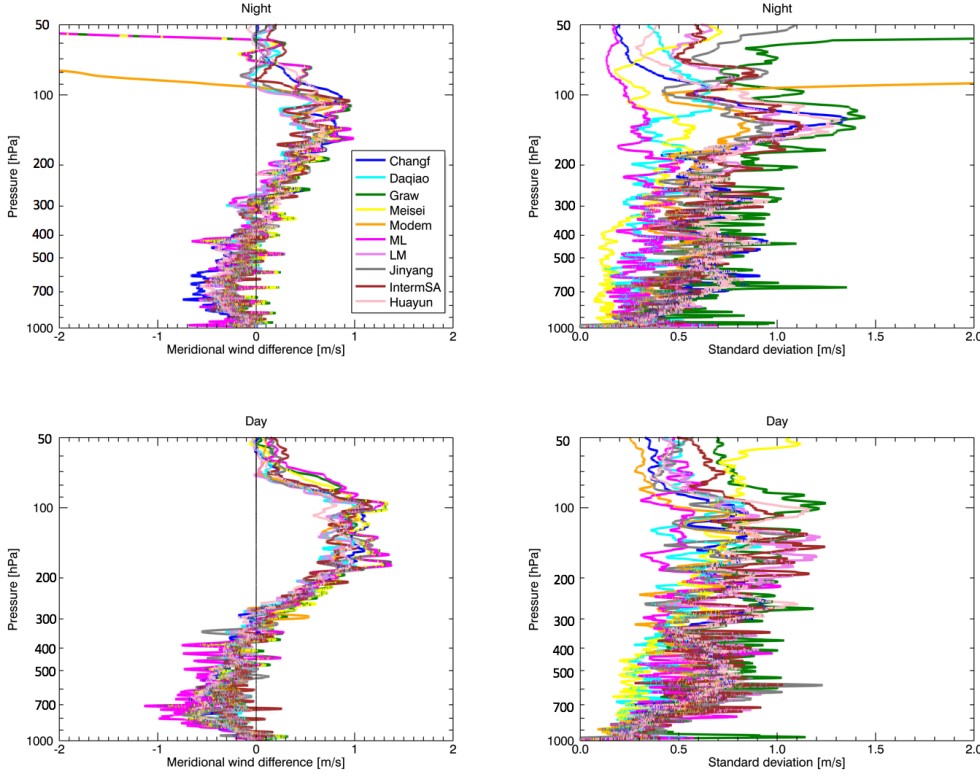

Figure 8: Same as Figure 5 but for meridional wind component (v).

Wind data provided with the RHARM approach must be used with caution considering that the
radiosonde types reported in Table 4 are processed with distinct software routines provided by the
respective manufacturers which apply distinct smoothing to the data. The unavailability of the raw
data does not enable reprocessing of the data to provide all of them at the same resolution or even
at a known resolution, which can be controlled for in the RHARM software and optimized to remove
spurious effects on the wind measurement by the radiosondes.

**4.   Results**
**4.1 RHARM consistency with GRUAN**
Although built to mimic the GDP, the RHARM the approach is not applied to the raw radiosonde
data. This may generate discrepancies in the result between the RHARM and the GDP which must
be quantified. By construction, the performance of the RHARM approach are expected be similar
on average to the GDP.



To evaluated the consistency of the RHARM adjustments applied to the RS92 IGRA sondes with the
GDP, in Figure 9 the "GRUAN minus RHARM'' mean difference profiles of temperature and RH are
compared with the corresponding profiles for "GRUAN minus IGRA". The plots in Figure 9 have been
limited to 100 hPa for the temperature and to 250 hPa for the RH: the latter is the minimum pressure
for the all the RH profiles adjusted using the RHARM approach, while for temperature adjustments
above 100 hPa are the same or very close to those carried out at 100 hPa. For temperature at night,
the difference GRUAN-IGRA is almost constant from the surface up to 300 hPa with a value of 0.12-
0.13 K, while below 300 hPa it is a slightly smaller with values of 0.1 K. Instead, the GRUAN-RHARM
difference is closer to zero along the entire pressure range with values smaller than 0.07 K up to 250
hPa and close to zero at higher altitudes. During the day, the GRUAN-IGRA difference is nearly
constant at all the pressure levels with a value of about 0.12 K. The standard deviation of the
difference is almost the same for night and day with increasing values towards lower pressures from
0.2 to 0.3 K. These values agree with the results of the comparison shown for GRUAN vs Vaisala data
products    (Dirksen    et    al.,    2014)    and    with    the    manufacturer    specifications
(https://www.vaisala.com/sites/default/files/documents/RS92SGP-Datasheet-B210358EN-F-
LOW.pdf). For at least some cases, the GRUAN-IGRA difference may be related to rounding of
temperature values in alphanumeric TEMP reports (Ingleby, 2017) and/or a systematic contribution
of 0.05 K due to the conversion of Celsius to Kelvin by the decoding software affecting alphanumeric
to BUFR transition when radiosoundings data are transmitted to the WMO Information System
(WIS).
For RH at night, the GRUAN-IGRA difference increases with height from less than 0.5% RH to 2.0%
RH and, during the day, from 0.7 % RH to 1.8% RH. The RHARM adjustments are able to reduce on
average the difference achieving negligible values, close to zero, both during night and day. The
standard deviation is similar for both the difference profile at night and day with values ranging
between 1.5 % RH and 5.0 % RH, increasing with decreasing pressures.
In analogy to temperature and RH, the wind speed mean differences have been calculated using
both night and daytime observations, because there is not any difference in the data processing
applied in the three considered datasets (GRUAN, IGRA and RHARM). Both the GRUAN-IGRA and
GRUAN-RHARM difference profiles, shown in Figure 10, are very close to zero from 1000 hPa to 300
hPa. Above this altitude, RHARM has a smaller mean difference than IGRA with respect to GRUAN
values, always positive and smaller than 0.05 m/s, while IGRA shows differences with GRUAN within
about ±0.3 m/s. The residual differences between GRUAN and RHARM may be due to several
reasons, such as rounding problems or differences in the smoothing window used by the
manufacturers and GRUAN data processing.

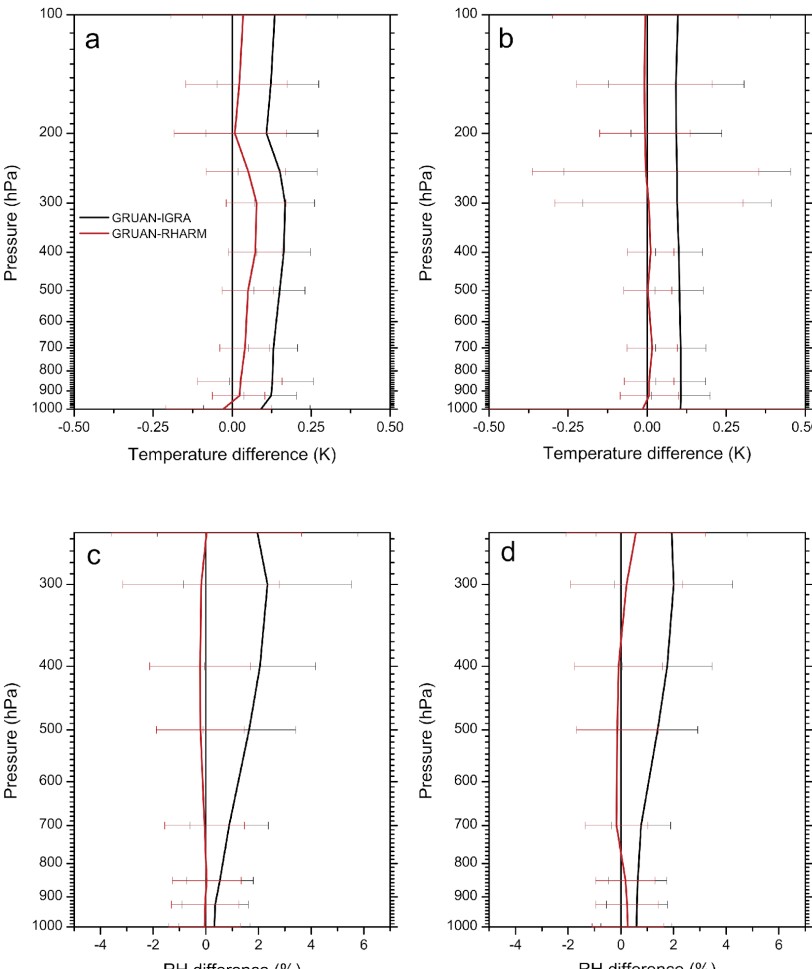

Figure 9: Mean difference profiles of temperature (top panels) and relative humidity (bottom panels) with the corresponding standard deviations (horizontal bar) calculated from the comparison of the night time (panels a and c) and daytime (panels b and d) difference "GRUAN minus IGRA" (black lines) and "GRUAN minus RHARM" (red lines) for the profiles available at all GRUAN stations, in the period 2010-2018.

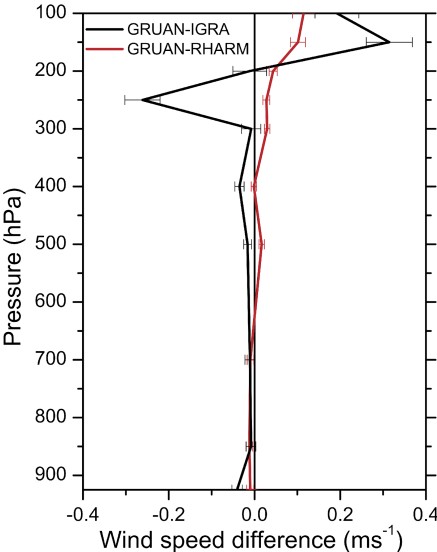

Figure 10: Same as panels in Figure 9 but for wind speed including both night and daytime observations.

In Figure 11, the probability density functions (pdfs) calculated for the IGRA and RHARM datasets
(Figure 1) in the Northern Hemisphere (NH) at 300 hPa are shown for temperature, RH and wind
speed components. The median, the first and third quartiles of the pdfs shown in Figure 11 are
reported in Table 5 for convenience. For temperature, it appears evident that the applied
adjustments minimally alter the IGRA pdf: the small magnitude of the RHARM adjustments for
temperature also indicates the enhanced quality of the data collected by most recent radiosonde
types available on the market compared to the historical observations (Thorne et al., 2012). The
RHARM pdf is slightly "warmer" than the IGRA one, with a median value 0.05 K larger, indicating
that an apparent systematic underestimation in the IGRA data.
For RH, there is a strong difference between the IGRA and RHARM pdfs mainly due to the
adjustment for the effects of solar radiation: the RHARM pdf is characterized by wetter values
revealing that the manufacturer applied correction is not sufficient and can provide too dry values.
The median value for RHARM is 2% larger than for IGRA: this result reflects the effect of the humidity
radiosonde dry-bias. RHARM has significant differences in its RH values compared to IGRA,
especially at RH values below 15% RH and above 55% RH.
For wind speed components, as anticipated, the systematic effects have a smaller magnitude than
for temperature and RH; the IGRA and RHARM pdfs are fairly similar with a difference of the median
value of about 0.1 ms$^{-1}$ for the u wind speed and of 0.52 ms$^{-1}$ for v, with the RHARM pdf more
skewed toward positive values than IGRA.
Figure 12 shows the same comparison as Figure 11 (300 hPa) but calculated for all stations in the
tropics (±25° latitude). The corresponding median, the first and third quartiles are reported in Table
6. Similar conclusions to Figure 11 can be drawn, in particular for temperature, as the warmer values
recorded in the NH by RHARM become more evident in the tropics (difference of 0.13 K in the
median). In general, the difference between IGRA and RHARM is the same as for NH: the
temperature pdf is closer to a normal distribution with much smaller variance, due to the larger
atmospheric stability and to the smaller seasonality, while the RH pdf is very similar to NH. The





difference in the RH median value is 2% RH like in the NH. The wind pdfs exhibit a difference in the
median values of 0.30 ms$^{-1}$ for the u wind speed and of 0.32 ms$^{-1}$ for v, with the RHARM pdf again
more skewed toward positive values than IGRA.

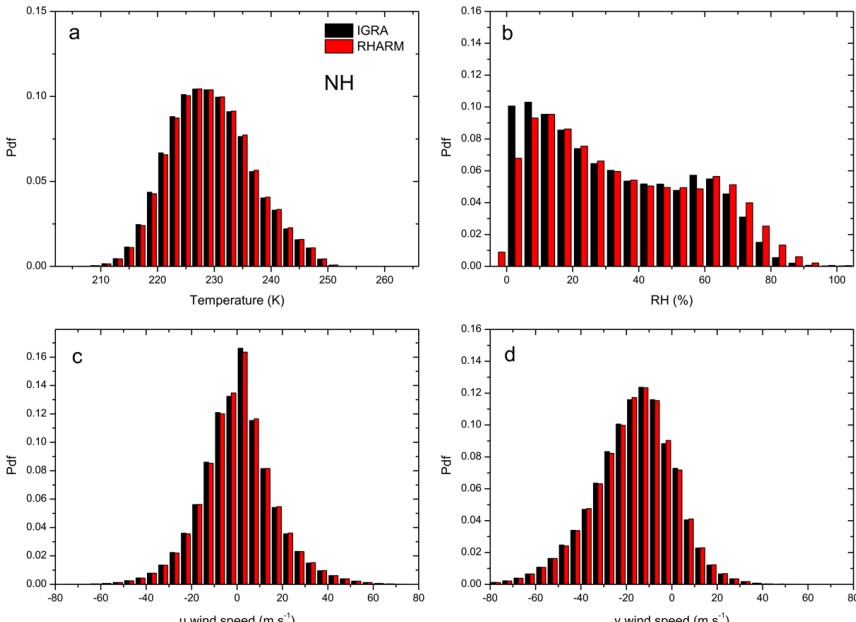

Figure 11: pdfs calculated in the Northern Hemisphere (NH) at 300 hPa for the IGRA and RHARM datasets of temperature
(panel a), RH (panel b), u wind component (panel c) and v wind component (panel d), using the station shown in Figure
738   1.

| NH | 1st Quartile (Q1) | Median | 3rd Quartile (Q3) |
|---|---|---|---|
| T IGRA (K) | 224.15 | 229.05 | 234.25 |
| T RHARM (K) | 224.22 | 229.10 | 234.27 |
| RH IGRA (%) | 12 | 28 | 51 |
| RH RHARM (%) | 14 | 30 | 54 |
| u IGRA (m s$^{-1}$) | -9.90 | 0.01 | 8.92 |
| u RHARM (m s$^{-1}$) | -9.05 | 0.11 | 9.26 |
| v IGRA (m s$^{-1}$) | -28.56 | -15.94 | -5.27 |
| v RHARM (m s$^{-1}$) | -27.46 | -15.42 | -4.94 |

Table 5: first, second (median) and third quartiles of the pdfs shown in Figure 11.


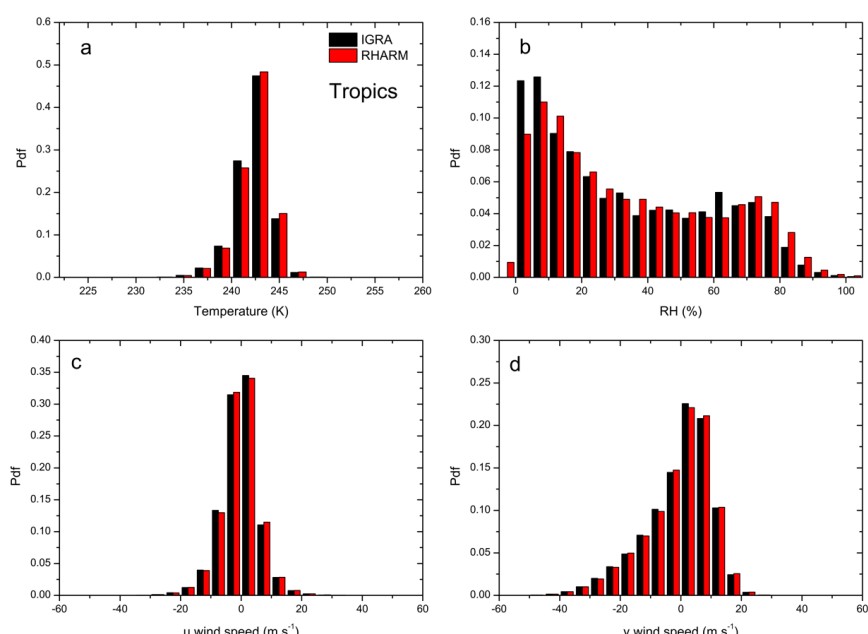

Figure 12: same as Figure 11 but calculated at the Tropics (±25° latitude).

| Tropics | 1st Quartile (Q1) | Median | 3rd Quartile (Q3) |
|---|---|---|---|
| T IGRA (K) | 241.25 | 242.45 | 243.45 |
| T RHARM (K) | 241.42 | 242.58 | 243.54 |
| RH IGRA (%) | 0.10 | 0.27 | 0.55 |
| RH RHARM (%) | 0.11 | 0.29 | 0.57 |
| u IGRA (m s$^{-1}$) | -4.11 | -0.50 | 2.92 |
| u RHARM (m s$^{-1}$) | -3.78 | -0.20 | 3.08 |
| v IGRA (m s$^{-1}$) | -7.46 | 1.23 | 6.80 |
| v RHARM (m s$^{-1}$) | -6.65 | 1.55 | 7.07 |

Table 6: first, second (median) and third quartiles of the pdfs shown in Figure 12.
In Figure 13, instead it is reported a comparison between the pdfs among the GRUAN, IGRA and
RHARM RH values for all the GRUAN stations in the period 2008-2018. The comparison comprises
all the night and daytime observations performed with the RS92 sondes on 00:00 and 12:00 UTC, at
300 hPa. The comparison between the two panels shows the impact of the RHARM adjustments
applied to the original IGRA data. The RHARM RH values become considerably more similar to
GRUAN, especially for values higher than 55% RH. These results imply that manufacturer data
processing applied to the RH radiosounding profiles measured by Vaisala RS92 radiosondes is not
adequate to compensate for instrumental effects, as it is inducing a dry-bias. Similar conclusions
can be inferred by the ID2010 data discussed above for the other radiosonde manufacturers.

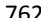

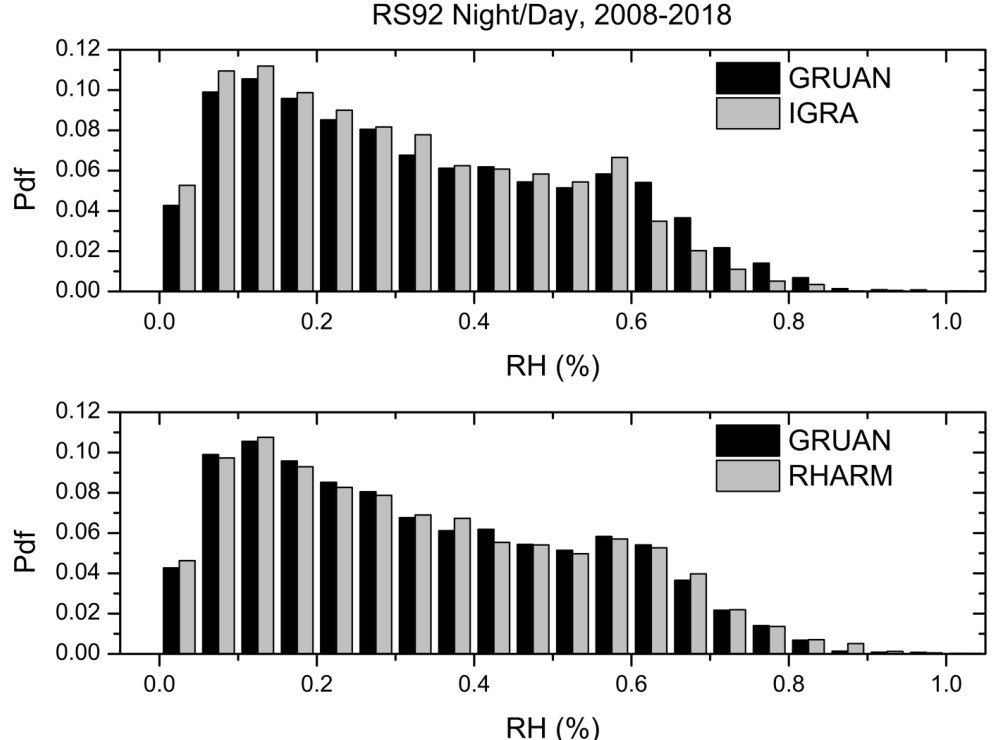

Figure 13: top panel, comparison between GRUAN (black) and IGRA (grey) RH measurements at 300 hPa for the profiles
available at all GRUAN stations (only RS92 sondes), in the period 2010-2018 The comparison comprises all the night and
daytime observations on 00:00 and 12:00 UTC. Bottom panel, same as top panel but for GRUAN (black) and RHARM
(grey).
**4.2 Comparisons with ERA5**
An important step in the performance assessment of the RHARM data is the comparison with
atmospheric reanalysis data. The latter incorporates millions of observations into a data assimilation
system, every 6-12 hours over the period being analyzed, providing a systematic approach to
produce data sets for climate monitoring and research. The various reanalysis products available
from the existing climate services have proven to be valuable when used appropriately (Dee et al.,
2016). Nevertheless, reanalysis reliability can considerably vary depending on the location, time
period, and variable considered (Dee et al., 2016). The changing mix of observations, and biases in
observations and models, can introduce spurious variability and trends into reanalysis output. In
this section, IGRA and RHARM are compared with the ERA5 ECMWF atmospheric reanalysis. ERA5
is the latest climate reanalysis produced by ECMWF providing hourly data on regular latitude-
longitude grids at 0.25° x 0.25° resolution (Hersbach et al., 2020), with atmospheric parameters on
37 pressure levels. ERA5 is publicly available through the Copernicus Climate Data Store (CDS,
https://cds.climate.copernicus.eu).
IGRA and RHARM monthly averages of temperature and RH have been compared with the monthly
averages obtained for the nearest ERA5 grid-point to each radiosounding station. Simultaneous
vertical profiles on 12 UTC and 00 UTC at mandatory levels have been considered only. Considering
the high resolution of ERA5 and its spatial representativeness, the representativeness uncertainty
due to the use of the nearest grid-point should be comparable with other methods (e.g. kriging,
bilinear interpolation, etc.).
Figure 14 compares the 300 hPa monthly zonal anomalies (i.e deviation from the mean created by
subtracting climatological values from monthly means) of temperature and of RH calculated
between 01/08/2006 and 01/08/2018 for IGRA, RHARM and ERA5 for Northern Hemisphere (NH),
tropics and Southern Hemisphere (SH) locations. Figure 15 shows the same as Figure 14 but for the
Arctic region (70°- 90° N) and the Antarctic region (70°- 90° S).

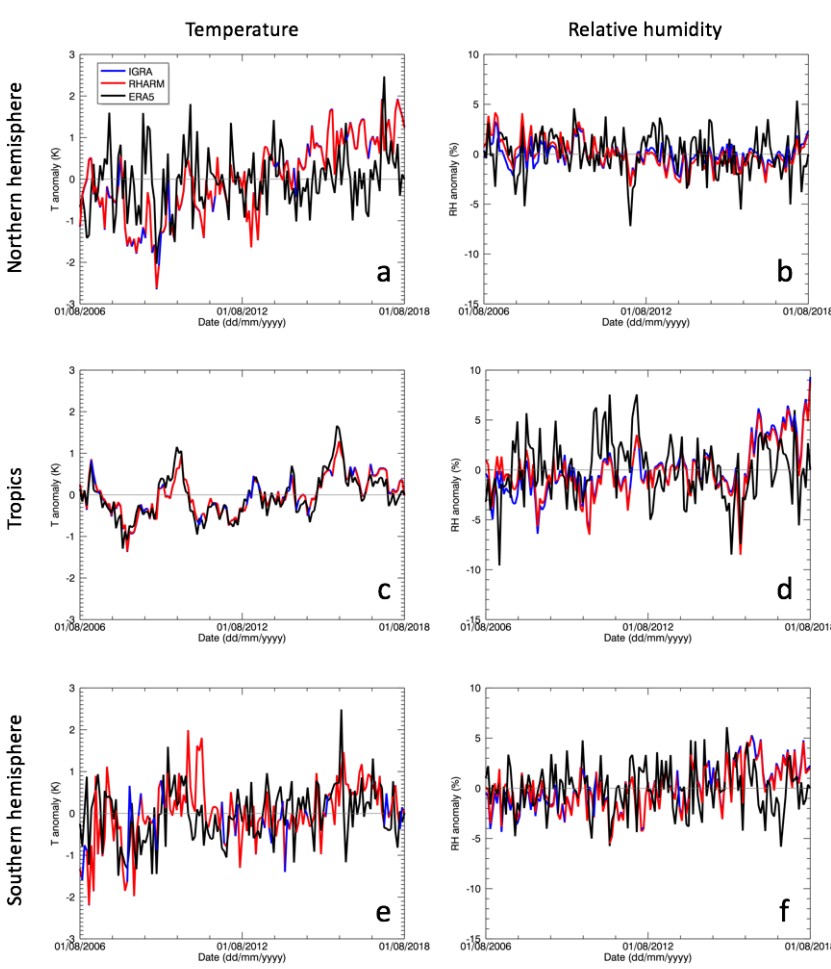

Figure 14: monthly temperature and RH anomalies calculated for IGRA (blue), RHARM (red) and ERA5 meteorological
reanalysis (black) at 300 hPa. Temperature anomalies are reported in the left panel, RH anomalies in the right panels.
Panel a and b are for NH, panel c and d for the tropics, panels e and f for SH.

In Table 7, the decadal trends for the time series of temperature monthly anomalies, shown in
Figure 14 and 15, are reported. Monthly anomalies are calculated aggregating all the available data





within each month, and each latitude region for both the observations and the reanalysis. Trends
are calculated on the monthly anomalies. Table 8, instead, refers to the decadal trends for the time
series of RH monthly anomalies. Both the tables also report the median absolute deviation (MAD)
from the fitted linear trends, which gives an estimate of the statistical uncertainty affecting the
trends. Trends have been calculated using a robust least absolute deviation method (Wong et al.,
1989). This method has proven to be equivalent to other regression methods commonly used in
literature, such Theil-Sen and Levenberg-Marquardt (Sy et al., 2020), though faster in terms of
computation efficiency.
For both T and RH, IGRA, RHARM and ERA5 show the same upward trend of 0.6 K/decade in the
tropics at 300 hPa, where also the agreement of the temperature anomalies is very good. Trends at
other latitudes are very close. In the NH (Figure 14a), at 300 hPa, a trend of 2.0 K/decade is
estimated from the observations while a trend of 0.5 K/decade is estimated from ERA5. Similar
results are obtained considering European stations only (Madonna, 2020). In the period 2007-2010,
the observations show negative anomalies like ERA5 but larger in absolute value. After 2015,
observed anomalies are positive and exhibit a greater upward trend than ERA5. RHARM values
slightly reduce the difference with ERA5. In Antarctica at 300 hPa (Figure 15c), ERA5 exhibits an
upward trend of 0.5 K/decade, which is in conflict with the almost zero-trend estimated for the
observations, although there is a good agreement between ERA5 and the observed anomalies after
822  2015.

For RH in the NH at 300 hPa (Figure 14b), the three datasets show a downward trend with a
maximum difference of 1.2%/decade. Similar results are obtained considering the European domain
only (Madonna, 2020). The RHARM anomaly is more positive than IGRA until 2012. In the tropics,
trends for RH show differences between observations (IGRA and RHARM) and reanalysis of up to
4%/decade. In the entire time series, differences in the anomalies are observed, the most prominent
after 2015 when, after a strong dry anomaly observed in 2013-2014 (-9% RH), an increasing positive
anomaly is observed ranging from 2% RH in 2015 to 6% RH in 2018. This observed positive anomaly
differs from the ERA5 values, which oscillate around zero in the same period, and it is independent
of the longitude (additional analysis not shown). Also in this case, RHARM adjustments tend to
slightly reduce the gap between IGRA and ERA5: the anomaly reduction is due to large adjustments
from RHARM applied in the period 2006-2008 enabling the removal of systematic effects on the
IGRA radiosounding profiles. This translates to a smaller trend in the period 2006-2018 for RHARM
RH time series than for IGRA. The strong positive humidity anomalies observed in the tropics appear
to be correlated with significant positive anomalies of the bi-monthly multivariate El Niño/Southern
Oscillation (ENSO) index "MEI.v"2" (Hu and Fedorov, 2017) available at
https://www.esrl.noaa.gov/psd/enso/mei) which start in January 2015 and reaches within the same
year values larger than 2.0. Boosted by an El Niño event, the year 2015 was the first of five
consecutive years among the six warmest years in the 140-year observational record (see
https://www.ncdc.noaa.gov/sotc/global) which may be related to the observed strong positive
anomalies of relative humidity at the tropics and in the SH. A possible positive trend in upper-
tropospheric humidity has been already claimed in previous work (e.g. Dessler and Davis, 2010).
In the SH, the situation is similar to the tropics with the following differences: the temperature
anomalies have a good agreement and show the same trends, although the values of the observed
extremes are much larger than in ERA5. For RH, a very similar scenario to the tropics is shown with
positive anomalies which starts in 2015 with values up 6% RH, also in this case not detected in ERA5.
For both the tropics and the SH, monthly anomalies at 500 hPa show a similar scenario to 300 hPa
(not shown), although with smaller differences among the datasets.





Finally, for both the Arctic and Antarctic (Figure 15), IGRA and RHARM show upward trends,
differently from ERA5, with a discrepancy smaller than 5%/decade in the Arctic and of 3%/decade
in the Antarctic.

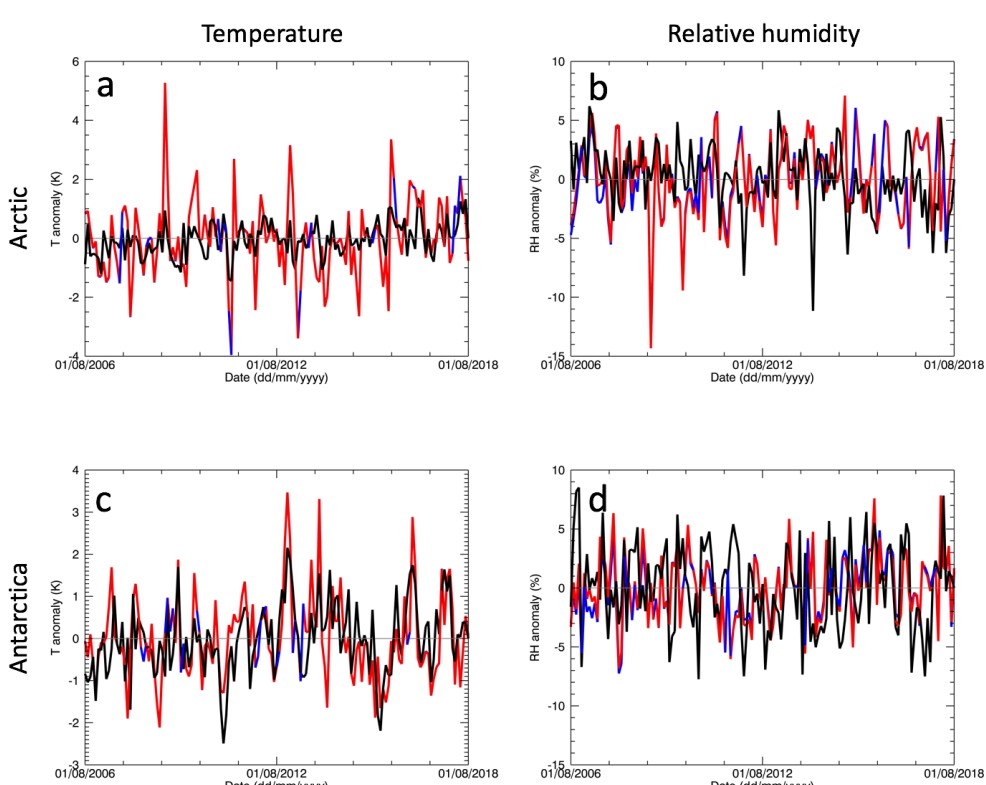

Figure 15: same as Figure 14 but panels a and b are for the Arctic, panel c and d for Antarctica.

| T trends (K/da) | IGRA trend | IGRA MAD | RHARM trend | RHARM MAD | ERA5 trend | ERA5 MAD |
|---|---|---|---|---|---|---|
| NH | 2.04 | 0.49 | 1.99 | 0.49 | 0.45 | 0.55 |
| Tropics | 0.60 | 0.32 | 0.56 | 0.32 | 0.57 | 0.38 |
| SH | 0.61 | 0.51 | 0.58 | 0.51 | 0.28 | 0.51 |
| Antarctic | -0.05 | 0.72 | -0.05 | 0.72 | 0.49 | 0.62 |
| Arctic | 0.43 | 0.93 | 0.41 | 0.93 | 0.58 | 0.42 |

Table 7: decadal trends of temperature (K/da) estimated using a robust least absolute deviation method for five zonal
regions using IGRA, RHARM, ERA5 data. For each dataset, two columns are reported in the table, one with the estimated
decadal trend and the other with the median absolute deviation (MAD) from the fitted linear trends.





| RH trends (%/da) | IGRA trend | IGRA MAD | RHARM trend | RHARM MAD | ERA5 trend | ERA5 MAD |
|---|---|---|---|---|---|---|
| NH | -0.3 | 2.74 | -1.5 | 2.92 | -0.6 | 5.05 |
| Tropics | 4.7 | 5.02 | 3.9 | 5.26 | 0.7 | 7.24 |
| SH | 3.9 | 4.29 | 2.9 | 4.62 | -0.1 | 5.66 |
| Antarctic | 2.5 | 6.39 | 1.5 | 6.87 | -0.4 | 9.13 |
| Arctic | 2.0 | 7.45 | 0.7 | 7.79 | -2.7 | 5.69 |

Table 8: same as Table 7 for RH.
**5. Uncertainties: consistency with GRUAN and independent validation**
A unique value of the RHARM dataset is that, for the first time, an estimation of the uncertainty is
provided for each single observation (i.e. at each pressure level). In this section, a statistical analysis
of the uncertainty values is provided. The differences between the RHARM approach and the GDP,
described in section 4.1, may also affect the quantification of the uncertainty budget, where the
unavailability of the raw radiosounding data forces the RHARM algorithm to use average statistics
to quantify a few uncertainty contributions instead of a point-by-point evaluation.
To investigate the consistency of the estimated values of the uncertainty by RHARM, Figure 16
undertakes a comparison between RHARM and GRUAN uncertainties for temperature and relative
humidity. The plots are based on the pdfs of the uncertainty estimated by using the data available
at sites reported in Table 1 and the corresponding observations from RHARM. Uncertainty for
RHARM is generally larger than the uncertainties obtainable using the GDP as expected given the
methodological considerations outlined in section 3. That is to say that the RHARM assumptions
increase, on average, the uncertainty compare to an ideally corresponding GDP.
In particular, for temperature (Figure 16, left panel), the median value of the GRUAN pdf is of 0.16
K versus a value of 0.22 K for RHARM (median values are considered for the analysis given the shape
of the pdf). The interquartile range (IQR) for GRUAN is 0.20 K while for RHARM is 0.26 K. These
numbers confirm that on average the uncertainty estimation obtained for RHARM overestimates
the GRUAN uncertainty. Nevertheless, due to the static nature of the assumptions made within
RHARM it might happen that the RHARM uncertainty may occasionally underestimate the GRUAN
uncertainty as occurs for a portion of values the below 0.1 K which increase the RHARM pdf
compared to GRUAN (Figure 16, left panel). These values are mainly related to night time
measurements.
For RH, the median value of the GRUAN pdf is of 1.1% versus a value of 3.6% for RHARM with an
IQR for GRUAN is 0.1 %, while for RHARM is 3.0 %. Maximum values observed with GRUAN are less
than 8 % while RHARM shows also values larger than 10 % and very few values larger than 20 %.
Finally, for the uncertainties of wind speed and direction, calculated using Eqs. 8 and 9, these are
based on the uncertainties calculated for u and v which, on their turn, are estimated as the addition
in quadrature of two constant terms, in opposition to GRUAN where the random uncertainty is
quantified at each measurement vertical level using a low-pass digital filter. The resulting typical
uncertainties for RHARM wind speed is 0.3-1.9 m s$^{-1}$ while for GRUAN is 0.1-1.3 m s$^{-1}$. For wind
direction, the typical values of the uncertainties are similar for both RHARM and GRUAN and in the
order of 1°.



The investigation of the pdf for the RHARM uncertainties as a function of latitudes for temperature
and RH (not shown) shows very similar shapes with small difference revealing, therefore, how the
uncertainties estimated by the RHARM approach are not latitude-dependent.

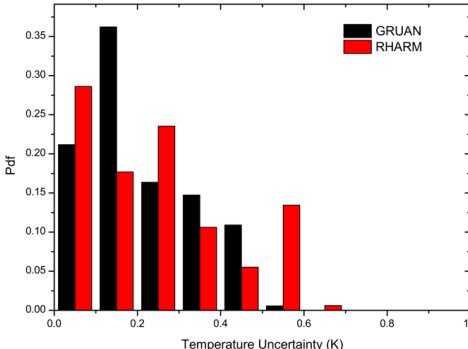 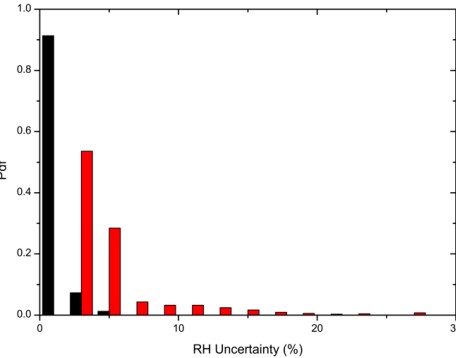

Figure 16: Comparison of pdfs of the uncertainty calculated using the GRUAN data processing (GDP) and the RHARM
approach at the six stations shown in Table 1. Pdfs are relative to temperature (panel a) and relative humidity (panel b)

To ascertain the quality of the estimated uncertainties, validation is an indispensable practice which
should be applied on every observational dataset. Validation of uncertainties means that these must
be "evaluated by independent means to establish quantitative realism and the credibility of the
uncertainty estimates" (Merchant et al., 2019). In order to provide a validation of the uncertainty
estimated by the RHARM approach, the methodology described in Merchant et al. (2019) has been
applied. This is based on the study of the probability density function of the ratio:

$$\frac{x_{RHARM}-x_{ref}}{\sqrt{u^2_{RHARM}+u^2_{ref}+u^2_{mis}}} \text{ [Eq. 16]},$$


where $x_{RHARM}$ is the measured estimate of the measurand, $x_{ref}$ indicates the estimate of the
measurand in the reference dataset used for the validation, u denotes the uncertainty and $u_{mis}$ is
the geophysical variability arising from temporal, spatial, and definitional mismatch between the
RHARM and reference data. A correct quantification of uncertainties and variability should be
reflected in a normal distribution of the ratio in Eq. 16, with a standard deviation equal to unity.

Acknowledging that the ideal solution for the validation must be based on independent reference
measurements (Thorne et al., 2017) of the same measurand, GRUAN data would be the ideal
candidate. However, RHARM has used information from and mimics part of the GDP meaning that
circularity considerations preclude its use for such a purpose. An alternative solution is adopted in
this paper which is to use the ERA5 background (6-hours forecast) as a reference value. Whilst this
background is a reliable estimation of the atmospheric state, it is not a real reference measurement
in that it is not itself an SI traceable measurement nor does it have comprehensive uncertainty
estimates. Observation minus Background departures have been already used as a diagnostic tool
for different latitude belts (Ingleby et al., 2017) because they can be considered, to a first
approximation, relatively homogenous. They also form the basis for the RAOBCORE / RICH family of
dataset approaches (Haimberger et al., 2012). Therefore, the use of the ERA5 background as a
reference for the test described in Eq. 16 appears to be a viable solution to infer quantitative





information for the validation of uncertainties. Other datasets may be used, such a GNSS-RO;
nevertheless, GNSS-RO are a valuable solution for dry temperatures in the UTLS, while for the mid
and lower troposphere the deconvolution of temperature and RH in the retrieval is dependent on a
first guess model. Furthermore, GNSS-RO rarely can provide profile information all the way down
to the surface.

Using the background as the reference dataset in Eq. 16, $u_{ref}$ has been estimated applying the Leave-
One-Out Cross validation method, LOOCV (Stone, 1974), to the background while $u_{mis}$ is evaluated
as the standard deviation of the O-B climatology at each station. In Figure 17, the ratio of Eq. 16 is
shown for O-B temperature and RH values for all the stations in the NH and in the tropics,
respectively, at 300 hPa. Each panel of Figure 17 also shows the best fitted normal distribution to
the data. O-B mean values are instead representative of O-B discrepancy. In the NH, the ratio for
temperature has a mean value of 0.1, while the standard deviation is 0.73 indicating that the
uncertainty at 300 hPa for temperature is overestimated of about 27%. The overestimated values
of the uncertainty increase the value of the pdf, compared to the fitted curve, in the middle of the
distribution while decreasing the pdf at the tails. In the tropics, a mean value of the ratio of -0.8 and
a standard deviation of 1.0 indicate that the uncertainty is well estimated with a small number of
overestimated values. For the RH, both in the NH and at the tropics, the uncertainty is
overestimated. For the NH, the mean value of the ratio is -1.3 and the standard deviation is 0.71
while in the tropics the mean value is -0.6 with a standard deviation of 0.84 and larger number of
overestimated values than in the NH. For RH uncertainties, the pdfs for both the NH and the tropics
are negatively skewed with a shape similar to the normal distribution except for an interval of values
on the right of the mean value. This might be related to systematic effects affecting the O-B
comparison, possibly due to inhomogeneities in the O-B departures within an entire latitude belt,
which can broaden the data O-B distribution and influence the value of the validation using the
model forecast as a reference.

In general, the RHARM uncertainties appears to be a good estimate or an overestimation of the
theoretical standard deviation. This can be considered a good result for the RHARM dataset
considering that dangerous underestimations of the uncertainties for temperature and RH values
can be considered a rare even. Nevertheless, the next version of the RHARM dataset will be
investigated to check whether the uncertainty overestimation, where occurring, could be reduced.
Future uncertainty assessments will be also oriented to the implementation of more sophisticated
models, using techniques like the kriging or modelling Gaussian processes to improve the capability
to estimate uref and umis to improve the characterization of the uncertainties in the RHARM
dataset.




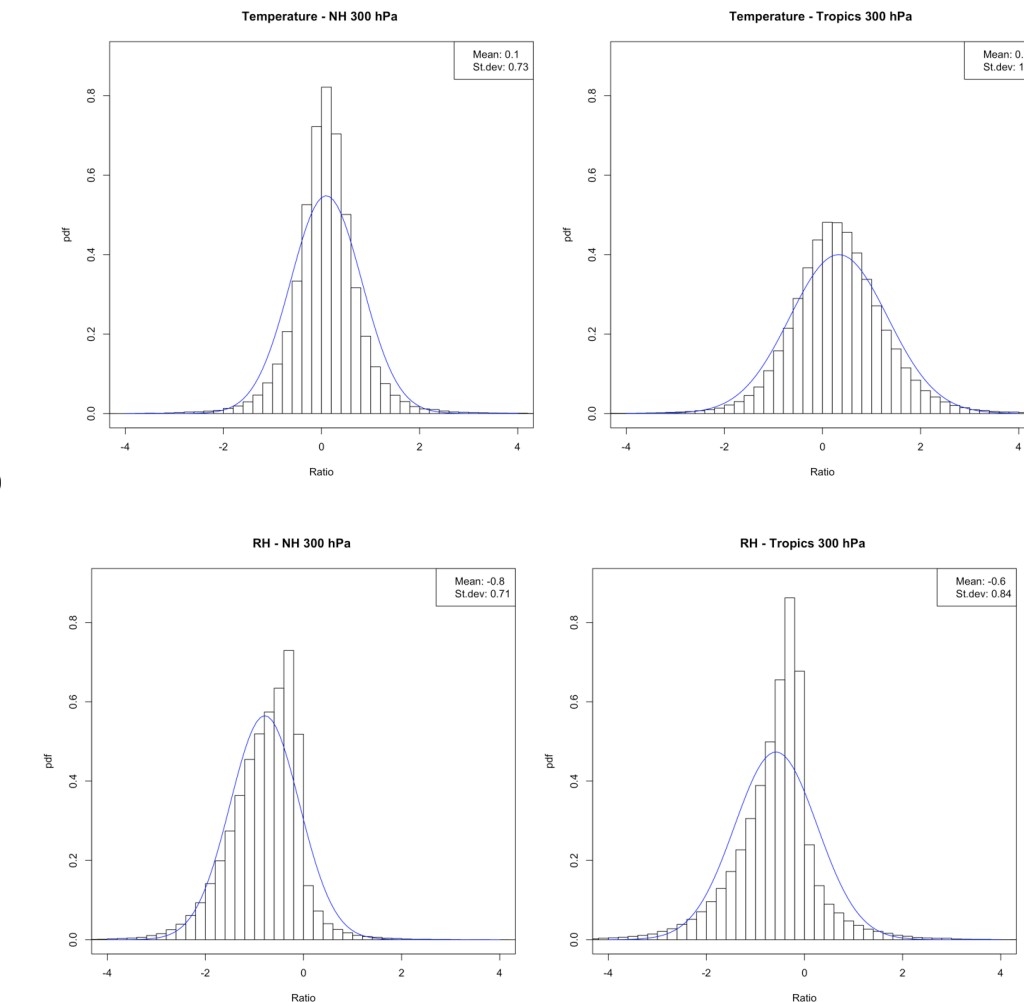



Figure 17: pdfs of the ratio reported in Eq. 16 calculated using O-B data (RHARM-minus-Background) in the NH (left panels) and in the tropics (bottom panels) at 300 hPa for temperature (top panels) and for RH (bottom panels) to validate the uncertainties estimated using the RHARM approach. Background data are from the ERA5 6-hours forecast model. For comparison with ideal uncertainty estimates, the best fitted normal distribution to each dataset (blue line) is also shown. In an ideal case where uncertainty would be properly estimated with the RHARM algorithm, the distribution should have a standard deviation equal to unity. Deviations from zero are due to the O-B discrepancy.

## 6. Conclusions and discussion

The work presented in this paper introduces the first metrologically-based component of the RHARM approach. RHARM is able to adjust a subset of historical radiosonde observations for which adequate metadata exist, and to quantify their uncertainties through a post processing chain based upon a combination of reference measurements provided by GRUAN and comparative performance measurements collected during the 2010 WMO/CIMO campaign.



The RHARM dataset provides one homogenization option, complementary to existing datasets of
homogenized radiosounding temperature measurements and to the handful of existing products
for RH and wind. RHARM differs from these previous efforts due to the use of "Reference
measurements" to calculate and adjust for systematic effects instead of using background
information provided by reanalysis, autoregressive models or neighboring stations. In addition, each
harmonized data series is provided with an estimation of the uncertainty. The different approach,
upon which RHARM is based, enables a more comprehensive exploration of structural uncertainties
in historical records.
In an ideal world, we would have access to the raw radiosounding data and be able to reprocess all
the data consistently to metrologically traceable standards. In the real world, save for GRUAN sites
and intercomparison campaigns, we do not have such an option. There is an action currently under
discussion by GCOS in its most recent Implementation Plan to explore the possibility to collect and
reprocess data from those sites who usually hold the original raw count data locally, although the
timeline and the resources to start the action are still uncertain.
The adjustments presented herein must be considered as the best solution/compromise between
the heterogeneity of the investigated manufacturer processed data profiles arising from IGRA and
the need to be coherent among the different adjustments calculated from the comparison with
different sources (e.g. GRUAN, ID2010).
The final goal of RHARM is to calculate average adjustments which should result in an improved
estimation of the climatological variability for temperature, humidity and wind profiles. This means
that on an individual station basis, the benefit of applying the proposed adjustment could be limited
or could even increase the difference with the "true" value or not properly estimate the uncertainty.
This is different from the solar radiation correction discussed in Section 3 which, though not exactly
the same as GDP, adjusts the data distribution, being applied as post-processing of the data and not
only as an average correction.
It is also very important to clarify that daytime corrections are representative of an average between
launches performed during the day at various local solar launch times and latitudes and, therefore,
various solar elevation angles. This can induce additional error sources which cannot be easily
quantified but which shall be considered and harmonized using statistical methods or inferred by
future radiosonde intercomparisons.
The RHARM algorithm also aims to show the importance of the availability of Reference data from
GRUAN and from the periodical WMO/CIMO radiosonde intercomparison data, as well as from
other experiments carried out according to the highest international best practices. These are
fundamental sources to quantify the uncertainties in the characterization of present and historical
radiosounding datasets. The collection and preservation of raw data by all radiosounding stations
would improve the basis to build the highest possible quality dataset of radiosounding
measurements. The future availability of new WMO/CIMO intercomparison data will enhance the
capability of the RHARM approach to improve the quality of both near-real time and historical
radiosoundings data. Moreover, the availability of the enhanced BUFR data reports (BTEM/BTEF
files replacing TEMP and previous BUFR version), for radiosounding measurement submitted to the
WIS, foster the reporting of high resolution vertical profiles with improved metadata, making the
gap between files reported by reference and baseline networks smaller. These files are made
available upon request by ECMWF (P.I. Bruce Ingleby) and will be processed and incorporated from
global observations shortly in the updated version of RHARM. The availability of metadata from
2016 on, when enhanced BUFR files start to be available, will also improve near real-time data
availability. In addition, the availability of new GRUAN data products, such as for the Meisei iMS-



100 sonde (Kobayashi et al., 2019), will be incorporated into subsequent versions of RHARM. These
innovations will further improve RHARM. The disagreement between the ERA5 reanalysis and the
observations anomalies, discussed in Section 4.2, are an example of the need to increase the
number and the quality of the observations available at the global scale with the estimation of the
related uncertainties.
In a follow-up paper, under preparation, an extension of the RHARM dataset to the historical
radiosounding time series will be presented. This extension starts from the RHARM post-processed
data, shown in this paper, used as an anchor point to homogenize radiosounding time series before
2004, at each single station, using statistical methods. In particular, the identification of change-
points in the time series is obtained applying a CUmulative SUMming (CUSUM) test while the
adjustments of instrumental effects are obtained adjusting iteratively the trend of the time series,
from the most recent data to the past (Madonna, 2020).
In conclusion, RHARM may initiate a new generation of homogenization techniques which fully
exploit the real value of reference measurements and of intercomparison datasets.

## 7. Data availability

A copy of the RHARM dataset is stored in the Copernicus Climate Data Store (CDS) although not
publicly available yet. For review purposes only, a subset has been made available at
http://doi.org/10.5281/zenodo.3973353 (Madonna et al., 2020a).

## 8. Acknowledgements

This work was done on behalf of the European Union's Copernicus Climate Change Service
implemented by ECMWF. Use of data as stated in the Copernicus license agreement is
acknowledged.
Thanks to the GRUAN Lead Center for sharing the Look-up table of the Streamer RTM. The Yangjiang
Intercomparison Dataset (ID2010) has been released upon agreement with the WMO YID protocol,
signed by CNR-IMAA and WMO on 27/07/2017.

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
