# Peer review of "Radiosounding HARMonization (RHARM): a new homogenized dataset of radiosounding temperature, humidity and wind profiles with uncertainty."

_Earth System Science Data, 2020_

## Referee Comment (RC1) · Anonymous Referee #1 · 23 Sep 2020

The authors tried to obtain a homogeneous data set of temperature, humidity and wind profile through developing a Radiosounding HARMonization (RHARM) approach. This approach first post-processes the data since 2004, and then detect and adjust systematic errors in the historical observations with the help of the documented metadata. This involves one important issue, that is, the stations used in the study have complete metadata records. How complete are the metadata records for the stations used? Incomplete metadata records will lead to many unadjusted jumps in the final data set, which can't be used for climate research and other applications, similar to the raw

dataset. Secondly, the data set derived from this study is mostly only since 2014 and lacks lots of stations in USA, Russia and China (Fig. 1) so as to prevent its applications in the future. Third, the verification of the derived data set is far from enough. This manuscript only shows the mean differences and PDF comparisons (whose differences are not obvious), which is not easy to see improvements from your approach. The authors will seek other ways to show the improvements of the derived data set. Fig. 14 shows time series that are not significantly different from the IGRA raw dataset and your homogenized dataset. Does this imply that the RHARM approach does not make significant adjustments for the raw data set? This reminds me that whether this small difference is due to incomplete metadata records, so that some jumps have not been adjusted? Lastly, the narrative structure of the piece and figure plots do not engender easy comprehension of claims, methods or main takeaways, which prevents a robust assessment. These may help improve the manuscript and requires extensive revisions.

Specific comments: 1. L21-25, do you mean the variables except relative humidity were harmonized on 16 standard pressure levels? Please clarify it. 2. L27-34, it is not clear that how to post-process, adjust and homogenize in each step? 3. L34, what is systematic effects? What's different from 'systematic biases or errors'? 4. L51, 'these climatic time series' is changed to 'these biased time series'? 5. L81, please cite other literatures at the end of the sentence and move (Hersbach et al., 2020) to the back of the ERA5? 6. L103, how many stations? instead of 'several sites' 7. L122, 'since 2004' is changed to 'from 2004'? 8. L132 and 179, pls revise 'a subset of 650 radiosounding stations'? 9. This study missed lots of stations in USA, Russia and China in Fig. 1? 10. Fig. 1 shows sonde types probably from different decades at different stations, which may mislead the readers. If there are several sonde types for one station, which type did you use? 11. L251, 'reasonably complete?' please see #8 12. L307, how did you handle surface data?

---

## Referee Comment (RC2) · Anonymous Referee #2 · 11 Oct 2020

Comments on "Radiosounding HARMonization (RHARM): a new homogenized dataset of radiosounding temperature, humidity and wind profiles with uncertainty" submitted to "Earth System Science Data" by Fabio Madonna et al.

The harmonization of radiosonde data recorded by different sensors under different conditions is no doubt a meaningful work for the scientific community and general users. The authors have processed a very huge dataset. As shown in Table 2, 2,205,200 radiosonde launches have been processed. The data processing workload in this paper is significant.

[Figure]

However I have a few major concerns regarding the paper.

1. The uncertainty equations such as eq. 5, eq. 7, as well as the ones in lines 472 and 473, are not mathematically derived based on the error propagation law. For instance, theoretically the eq. 5 should be derived from the eq. 4 based on error propagation law. It is unknown how the authors get the eq. 5.

2. The quantization of the variance of each term in eq. 5 as well as other equations is not clearly stated. How to estimate the final, total values of the uncertainties of temperature, relative humidity and wind? It is not clearly explained.

3. In line 175, it stated that "IGRA contains observations from several networks and initiatives, including the GCOS Upper-air Network (GUAN), and the universal RAwin-sonde OBservation program (RAOB)." So GUAN data (actually the correct name should be GRUAN data) is a part of the IGRA data. You processed data from 650 stations and all the data were also from the IGRA. In the Section 4.1 "RHARM consistency with GRUAN", the RHARM results are compared with GRUAN result. So authors are regarding the GRUAN data as reference data. This raises one new question. Your RHARM results are derived from GRUAN data and now your results are compared with the GRUAN data. It is a repeated use of the same dataset. I am not sure how you can get an objective assessment of your RHARM result.

In the Section 4.1 "RHARM consistency with GRUAN", though the title of this section is the "consistency with GRUAN". However in your Figures 9-13, you also compared your results with the IGRA data. As said earlier, IGRA data set include the GRUAN data set and RAOB data set. So it is confusing to readers: Your results are compared to GRUAN data, and are also compared to IGRA data.

4. In the Section 4.2 "Comparisons with ERA5", you compared your RHARM results with ECMWF ERA5 results. The derivation of ECMWF ERA5 results have used the global radiosonde datasets already. Therefore your comparison with ERA5 is not an independent assessment either. You repeatedly use the same set of radiosonde data.

5. In the section 5 "Uncertainties: consistency with GRUAN and independent validation", the GRUAN datasets are compared again here. It seems to be a repetition of the work in section 4.1. In Fig. 16, only 6 stations are used in the comparison. I am not sure how useful/meaningful it is to compare with only 6 stations' result, considering you are processing global 650 stations.

6. The conclusion is too long and you should summarize the main findings of the paper. In addition, there is no single numerical value to show your findings, which is surprise to me.

Other minor comments are:

In the paper, in some places it claims it has processed the historical data since 1978. However in some places it states it processed the data since 2004 to present and the 2010 WMO/CIMO radiosonde data. It is confusing to readers which dataset you have processed exactly.

In some occasions, abbreviations are not fully spelled in their first appearance. In some cases, the abbreviations are defined or redefined after its first use.

Lines 187-189, It stated "Beyond the 650 homogenized stations, also the other radiosounding profiles available from IGRA with documented metadata and a radiosonde model compatible with the GDP or the ID2010 have been post-processed using RHARM." It is very confusing to readers. How many stations did you do the RHARM processing, 650 stations or 650 stations plus some other radiosounding profiles available from IGRA? Why don't you give an exact value of the total number of stations you have processed?

Line 253, "The station density in North America, North East Asia, and East Africa is lower than in Europe, U.S and South America." This statement needs to be revised. North America includes U.S. It is not logical to compare U.S. with U.S.

Line 255, It stated "while the stations with largest number of launches are quite uniformly distributed globally (Figure 2)". This is not true. The red dots (I think they represent the stations with largest number of launches) are not uniformly distributed globally. The Figure 2 clearly show that Europe has the highest concentration.

Line 275, "GCOS Upper Air Network (GUAN) sites" Do you mean "GCOS Reference Upper Air Network (GUAN) sites"? In many places the GRUAN is spelled as GUAN.

The paper is very long and not easy to understand. Some sections contain too many contents and increases the difficulty to comprehend. Some sentences run a few lines and also reduce the readability.

Some tables/figures can be combined to increase readability. For instance, table 7 and table 8 can be combined.

---

## Author Comment (AC1) · 6 Nov 2020

The harmonization of radiosonde data recorded by different sensors under different conditions is no doubt a meaningful work for the scientific community and general users. The authors have processed a very huge dataset. As shown in Table 2, 2,205,200 radiosonde launches have been processed. The data processing workload in this paper is significant.

**We have examined in detail the comments provided by the anonymous referee #2 and thank the reviewer for the positive comments related to the amount of work undertaken and acknowledge his/her effort in providing comments aimed at improving the clarity and the robustness of the manuscript.**

**Below, we give in bold our replies to the reviewer's comments**

However, I have a few major concerns regarding the paper.
1. The uncertainty equations such as eq. 5, eq. 7, as well as the ones in lines 472 and 473, are not mathematically derived based on the error propagation law. For instance, theoretically the eq. 5 should be derived from the eq. 4 based on error propagation law. It is unknown how the authors get the eq. 5.

**For the treatment of errors and uncertainties we follow the approach outlined in the "Guide to the expression of uncertainty in measurement" by the working group 1 of the Joint Committee for Guides in Metrology (JCGM/WG 1, 2018).**
**"The uncertainty is an estimate of the standard deviation and characterizes the range of values (dispersion) that can be expected for a measurement.**
**Combined standard uncertainty: standard uncertainty of the result of a measurement when that result is obtained from the values of a number of other quantities, equal to the positive square root of a sum of terms, the terms being the variances or covariances of these other quantities weighted according to how the measurement result varies with changes in these quantities.'**
**Combined standard uncertainty may contain terms whose components are derived from Type A and/ or Type B evaluations without discrimination between types".**

**The uncertainty equations such as eq. 5, eq. 7 combine the various sources of uncertainties as prescribed to estimate the total uncertainty. The considered sources of uncertainties are both random and systematic. The majority of the corrections applied in RHARM processing aim to reduce the systematic effects in the measurements. The total uncertainty is the combined standard uncertainty equal to the positive square root of the sum of the squares of all the uncertainty contributions.**

**This approach ensures the greatest possible consistency with the uncertainty estimation provided by the GRUAN Data Processing.**

**With the aim to increase the clarity of the provide text and formulas to explain the calculation of the uncertainties, lines 376-389 have been modified as follows: "The combined standard uncertainty on $T_{RHARM,RS92}$, $\varepsilon(T_{RHARM,RS92})$, is calculated as follows:**

$$\varepsilon\left(T_{RHARM,RS92}\right) = \left(\sum_i \varepsilon^i_{systematic}(\Delta T)^2 + \varepsilon_R(\Delta T)^2\right)^{\frac{1}{2}} = \left(\varepsilon_{c,I_a}(\Delta T)^2 + \right.$$

$$\left. \varepsilon_{c,R_c}(\Delta T)^2 + \varepsilon_{vent}(\Delta T)^2 + \varepsilon_r(\Delta T)^2 + \varepsilon_R(\Delta T)^2\right)^{\frac{1}{2}} \textit{[Eq.5]}$$

*In Eq. 5, $\varepsilon^i_{systematic}(\Delta T)$ indicates a systematic uncertainty contribution; $\varepsilon_{c,I_a}(\Delta T)$ is the uncertainty due to the estimation of the solar actinic flux; $\varepsilon_{c,R_c}(\Delta T)$ is the uncertainty due to parameters estimated in the radiation correction model reported in Eq. 1. Formulas to calculate $\varepsilon_{c,I_a}(\Delta T)$ and $\varepsilon_{c,R_c}(\Delta T)$ are fully documented in Dirksen et al. (2014). $\varepsilon_{vent}$ is the uncertainty due to the ventilation rate (including the effect of the pendulum motion of the radiosonde assumed as in GRUAN to be of about 0.2 m s$^{-1}$); $\varepsilon_r$ indicates the comparison uncertainties estimated from the standard deviation of $\Delta T_r$. In RHARM, $\varepsilon_R$ is the random uncertainty with a fixed value of 0.15 K chosen in agreement with the GDP approach (Dirksen et al., 2014). When the radiation correction of the manufacturer is left unchanged, $\varepsilon\left(T_{RHARM,RS92}\right)$ is assumed to be the same as the closest temperature profile in time measured under the same meteorological conditions (i.e. clear sky or cloudy)."*

2. The quantization of the variance of each term in eq. 5 as well as other equations is not clearly stated. How to estimate the final, total values of the uncertainties of temperature, relative humidity and wind? It is not clearly explained.

**We have provided the detail of each of the uncertainty sources in the equation describing the calculation of the combined standard uncertainty for temperature, RH and wind.**
**In details for temperature, please see the previous comment.**

**For RH, at lines 430-437, the text described the calculation of the combined standard uncertainty for RH:** *"Likewise Eq.5, $\varepsilon\left(T_{RHARM,RS92}\right)$, the combined standard uncertainty on $RH_{RHARM,RS92}$, $\varepsilon\left(RH_{RHARM,RS92}\right)$, is calculated as follows:*

$$\varepsilon\left(RH_{RHARM,RS92}\right) = \left(\varepsilon_{RC_T}(\Delta RH)^2 + \varepsilon_{RC_f}(\Delta RH)^2 + \varepsilon_{cf}(\Delta RH)^2 + \varepsilon_R(\Delta RH)^2\right)^{\frac{1}{2}} \textit{[Eq.7]}$$

*Where $\varepsilon_{RC_T}(\Delta RH)$ is the uncertainty of dry bias correction; $\varepsilon_{RC_f}(\Delta RH)$ is the uncertainty of the radiation sensitivity factor f in Eq. 5; $\varepsilon_{cf}$ is the uncertainty due to calibration factor cf; $\varepsilon_R$ is an additional random uncertainty of 2% RH. In analogy with temperature, when the radiation correction of the manufacturer is left unchanged, $\varepsilon\left(RH_{RHARM,RS92}\right)$ is assumed to be the same as the closest RH profile in time measured under the same meteorological conditions."*

**At lines 490-497 report the formula for the estimation of uncertainties for the wind related quantities:** *"To adjust the IGRA wind profiles, the day and night time differences for u and v between the GRUAN processed and the IGRA radiosounding wind profiles have been calculated using the stations in Table 1. The approach is the same as for temperature (Eq.4), although it is reduced to $\Delta u_{RHARM,RS92} = \Delta u_r$ and to $\Delta v_{RHARM,RS92} = \Delta v_r$, for each of the wind vectorial components. The standard deviation of the $\Delta u_{RHARM,RS92}$ and $\Delta v_{RHARM,RS92}$ are then used as the estimation of the combined standard uncertainties, which will be expressed as $\varepsilon\left(\Delta u_{RHARM,RS92}\right) = \left(\varepsilon_r(\Delta u)^2 + \varepsilon_R(\Delta u)^2\right)^{\frac{1}{2}}$ and $\varepsilon\left(\Delta v_{RHARM,RS92}\right) = \left(\varepsilon_r(\Delta v)^2 + \varepsilon_R(\Delta v)^2\right)^{\frac{1}{2}}$. $\varepsilon_R$ is a random uncertainty of 0.15 m s$^{-1}$ for both u and v. "*
**Finally, for the adjustments derived from the ID2010 dataset (2010 WMO/CIMO radiosonde intercomparison), the calculation of uncertainties, already described in the manuscript, has been further clarified.**

3. In line 175, it stated that "IGRA contains observations from several networks and initiatives, including the GCOS Upper-air Network (GUAN), and the universal RAwinsonde OBservation program (RAOB)." So, GUAN data (actually the correct name should be GRUAN data) is a part of the IGRA data. You processed data from 650 stations and all the data were also from the IGRA. In the Section 4.1 "RHARM consistency with GRUAN", the RHARM results are compared with GRUAN result. So authors are regarding the GRUAN data as reference data. This raises one new question. Your RHARM results are derived from GRUAN data and now your results are compared with the GRUAN data. It is a repeated use of the same dataset. I am not sure how you can get an objective assessment of your RHARM result.

In the Section 4.1 "RHARM consistency with GRUAN", though the title of this section is the "consistency with GRUAN". However, in your Figures 9-13, you also compared your results with the IGRA data. As said earlier, IGRA data set include the GRUAN data set and RAOB data set. So, it is confusing to readers: Your results are compared to GRUAN data, and are also compared to IGRA data.

**The difference between GRUAN and GUAN has been clarified above, and it is fully documented in the manuscript. GUAN data are part of IGRA but GRUAN data not: GRUAN data are fully traceable radiosounding profiles with quantified uncertainties which is completely different from any other existing radiosounding dataset. Other existing radiosonde data archives are essentially based on the manufacturer data processing and do not provide any estimation of the uncertainty. Therefore, GRUAN data are not contained in the 691 RHARM processed stations although data from the sonde processed using the manufacturer processing software do indeed form part of the iGRA database.**

**The RHARM approach is inspired by the GRUAN data processing, although uncertainties in RHARM cannot be quantified exactly as done by GRUAN which requires the raw digital counts as the basis for the processing. The results reported in section 4.1 ("RHARM consistency with GRUAN") are not meant to be an objective assessment of the RHARM dataset, but they quantify the residual gaps between RHARM and GRUAN resulting from the lack of raw data availability and the necessary compromises this entails at the vast majority of global radiosonde sites.**

**Regarding Figures 9-13, the reviewer is right, a comparison with IGRA data is included in the section entitled "RHARM consistency with GRUAN". In the new version of the manuscript, a new subsection has been created to separate the comparison between IGRA and RHARM data.**

4. In the Section 4.2 "Comparisons with ERA5", you compared your RHARM results with ECMWF ERA5 results. The derivation of ECMWF ERA5 results have used the global radiosonde datasets already. Therefore, your comparison with ERA5 is not an independent assessment either. You repeatedly use the same set of radiosonde data.

**Theoretically, we agree with the referee. This is the reasons why the authors introduced the comparison discussed in section 4.2 as "An important step in the performance assessment of the RHARM data" and not as an "independent assessment". Despite a degree of non-independence due to use of global radiosoundings data for anchoring its bias correction, it is also true that ERA5 reanalysis assimilation scheme selects the radiosonde stations using quality criteria which are completely different from RHARM and permits the ingestion of a different, typically smaller, set of radiosounding observations.**

**Furthermore, in the period under investigation ERA5 benefits from ingest of passive satellite retrievals, GNSS-RO and commercial aircraft so in many locations the radiosonde influence on ERA5 is greatly diminished. For the reported comparison, ERA5 applies an adaptive bias correction to avoid systematic effects due to the biases affecting radiosounding measurements in the upper troposphere/lower stratosphere.**

**In comparison, RHARM applies a completely different approach to adjust the instrumental bias. Moreover, there are several papers in literature and technical documents from ECMWF reporting differences between the ERA5 and the radiosounding data (https://www.ecmwf.int/sites/default/files/elibrary/2020/19362-global-stratospheric-temperature-bias-and-other-stratospheric-aspects-era5-and-era51.pdf; Zhou, Q.; Zhang, Y.; Jia, S.; Jin, J.; Lv, S.; Li, Y. Climatology of Cloud Vertical Structures from Long-Term High-Resolution Radiosonde Measurements in Beijing. Atmosphere 2020, 11, 401). This is due to the nature of the reanalysis data assimilation system and to the huge amount of ingested data sources.**

**It is also important to point out that the comparison of homogenized datasets with reanalysis is quite often adopted in literature and this is due to the fact that reanalysis is heavily used in climate studies.**

**The optimal basis for a comparison would be to use ERA5 reanalysis data or some other data source obtained without the ingestion of radiosounding measurements and that thus was completely independent. Given the need for a vertically resolved profile only GNSS-RO may come close. A perfectly independent assessment of the RHARM data can be carried out using Cryogenic Frostpoint Hygrometer (CFH) data, but these are available with long records only at a very few sites.**

**However, in order to more clearly show the improvements of the RHARM dataset, also following on from comments by referee #1, two further comparisons have been added to section 5. Firstly, to show the improvement brought by the RHARM dataset to radiosounding temperature data, the episodes of 2017 and 2018 Sudden Stratospheric Warming (SSW) events are shown using IGRA, RHARM and ERA5 data, while for relative humidity a comparison with MLS-AURA data at 300 hPa is discussed. These additional comparisons are reported in an appendix of the new version of the manuscript and summarized in the reply to the referee #1 (to avoid repetition the same content is not reported here as well).**

5. In the section 5 "Uncertainties: consistency with GRUAN and independent validation", the GRUAN datasets are compared again here. It seems to be a repetition of the work in section 4.1. In Fig. 16, only 6 stations are used in the comparison. I am not sure how useful/meaningful it is to compare with only 6 stations' result, considering you are processing global 650 stations.

**Please see the comments to the point 3, also considering that an independent validation of the uncertainties is provided in section 6 at a much broader scale.**

6. The conclusion is too long and you should summarize the main findings of the paper. In addition, there is no single numerical value to show your findings, which is surprise to me.

**Conclusions have been modified using a more schematic style with a clearer description of the main quantitative results discussed in the manuscript.**

Other minor comments are:
In the paper, in some places it claims it has processed the historical data since 1978. However, in some places it states it processed the data since 2004 to present and the 2010 WMO/CIMO radiosonde data. It is confusing to readers which dataset you have processed exactly.

**In the new version of the manuscript, a clear difference was made between what is discussed in this manuscript (adjustment of radiosondes from 2004) and what will be available to the users in the final RHARM dataset (adjustment of radiosondes from 2004 plus statistical homogenization of radiosonde from 1978 to 2004). A second manuscript currently under preparation will complete the description of the RHARM approach.**

**To this purpose lines 115-133 of the manuscript discussion version have been modified as follows:**
*"Under the Copernicus Climate Change Service (C3S) activities, a novel algorithm has been designed and implemented for homogenization of historical radiosounding data records available since 1978 (earlier records are not assessed due to the more heterogeneous data availability at mandatory levels before). The approach is named RHARM (Radiosounding HARMonization) and is based on two main steps:*

  a. *Adjustment of systematic effects and quantification of uncertainties by adjusting the radiosounding observations of temperature, humidity and wind from 2004 to present using the GRUAN data and algorithms as well as the 2010 WMO/CIMO radiosonde intercomparison dataset [hereinafter ID2010, Nash et al. 2011], made available upon agreement with WMO;*

  b. *Identification of change-points in the earlier portions of the time series and adjustment of non-climatic (systematic) effects using statistical methods with related quantification of uncertainties in the historical observations.*

*The present paper provides an analytic description of the section of the RHARM approach described in the point a above and able to adjust a subset of 691 radiosounding stations available from the Integrated Global Radiosonde Archive (IGRA - Durré et al., 2006; Durré et al., 2018) from 2004. "*

**Therefore, we are discussing the section of the RHARM algorithm obtained by adjusting the RS92 Vaisala radiosondes and all the other radiosonde types involved in WMO-CIMO 2010 radiosonde intercomparison. The identification of these sonde type within IGRA was based on the metadata collected by IGRA itself (i.e. the radiosonde code related to the WMO 3685 table) and on the use of the TAC code made available in the more recent high-resolution BUFR files provided by an increasing number of stations (about 1-2 hundreds at present) and supplied for RHARM by the ECWMF; the files includes extensive metadata and a larger number of pressure levels for each reported radiosounding launch. The selected radiosondes types are generally available from 2004, with differences in the covered period depending on each single station.**
**The homogenization of the historical data before 2004 is alluded to in this paper to create the link to the second paper describing the RHARM algorithm, in preparation.**

In some occasions, abbreviations are not fully spelled in their first appearance. In some cases, the abbreviations are defined or redefined after its first use.

**In the new version of the manuscript, we provide the spelling for all the abbreviations and redundancies have been removed.**

Lines 187-189, It stated "Beyond the 650 homogenized stations, also the other radiosounding profiles available from IGRA with documented metadata and a radiosonde model compatible with the GDP or the ID2010 have been post-processed using RHARM." It is very confusing to readers. How many stations did you do the RHARM processing, 650 stations or 650 stations plus some other radiosounding profiles available from IGRA? Why don't you give an exact value of the total number of stations you have processed?

**In the new version of the manuscript, the exact value of the total number of processed stations has been reported.**

Line 253, "The station density in North America, North East Asia, and East Africa is lower than in Europe, U.S and South America." This statement needs to be revised. North America includes U.S. It is not logical to compare U.S. with U.S.

**This is fixed in the new version of the manuscript.**

Line 255, It stated "while the stations with largest number of launches are quite uniformly distributed globally (Figure 2)". This is not true. The red dots (I think they represent the stations with largest number of launches) are not uniformly distributed globally. The Figure 2 clearly show that Europe has the highest concentration.

**The sentence has been revised as follows:** *"The station density in North America, North East Asia, and East Africa is lower than in Europe, U.S and South America, but this is common to all such products and reflects historical observing system inadequacies."*

Line 275, "GCOS Upper Air Network (GUAN) sites" Do you mean "GCOS Reference Upper Air Network (GUAN) sites"? In many places the GRUAN is spelled as GUAN.

**GUAN and GRUAN are different networks: As described in the manuscript, GRUAN is the upper-air reference network (www.gruan.org) while GUAN is the upper-air baseline component of the GCOS (https://gcos.wmo.int/en/networks/atmospheric/guan). However, we have revised the manuscript to ensure that the two have been used properly throughout.**

The paper is very long and not easy to understand. Some sections contain too many contents and increases the difficulty to comprehend. Some sentences run a few lines and also reduce the readability.
Some tables/figures can be combined to increase readability. For instance, table 7 and table 8 can be combined.

**Table 7 and 8 have been combined. Nevertheless, in the new version of the manuscript we have tried to improve the text removing unneeded sentences but have had to find a trade-off between**

the referee's comments. In particular, the referee #1 asked to extend the section where the verification of the improvements achieved by the RHARM approach is discussed.

In our opinion, the paper, as a dataset description paper, must be very analytic to guide the users toward the full exploitation of the dataset itself. This also means that the paper has to be sufficiently long to ensure all the details are provided to the reader, along with all the needed demonstrations that can increase confidence in the RHARM data usage.

---

## Author Comment (AC2) · 6 Nov 2020

Reply to the interactive comment by the Anonymous Referee #1 on "Radiosounding HARMonization (RHARM): a new homogenized dataset of radiosounding temperature, humidity and wind profiles with uncertainty" by Fabio Madonna et al.

**The authors of the manuscript by F. Madonna et al examined in detail the comments provided by the anonymous referee #1 and acknowledge his/her effort in providing comments aimed at improving the robustness of the manuscript.**

Below, the authors report in bold their replies to the reviewer's comments

The authors tried to obtain a homogeneous data set of temperature, humidity and wind profile through developing a Radiosounding HARMonization (RHARM) approach. This approach first post-processes the data since 2004, and then detect and adjust systematic errors in the historical observations with the help of the documented metadata. This involves one important issue, that is, the stations used in the study have complete metadata records. How complete are the metadata records for the stations used? Incomplete metadata records will lead to many unadjusted jumps in the final data set, which can't be used for climate research and other applications, similar to the raw dataset.

As described at lines 115-129 of the manuscript discussion version "In the frame of the Copernicus Climate Change Service (C3S) activities, a novel algorithm has been designed and implemented for homogenization of historical radiosounding data records available since 1978 (earlier records are not assessed due to the more heterogeneous data availability at mandatory levels before). The approach is named RHARM (Radiosounding HARMonization) and it is based on two main steps:

a. Adjustment of systematic effects and quantification of uncertainties by adjusting the radiosounding observations of temperature, humidity and wind from 2004 to present using the GRUAN data and algorithms as well as the 2010 WMO/CIMO radiosonde intercomparison dataset [hereinafter ID2010, Nash et al. 2011], made available upon agreement with WMO;

b. Identification of change-points in the time series and adjustment of non-climatic (systematic) effects using statistical methods with related quantification of uncertainties in the historical observations."

Therefore, in the present manuscript we are discussing the section of the RHARM algorithm obtained by adjusting the RS92 Vaisala radiosondes and all the other radiosonde types involved in WMO-CIMO 2010 radiosonde intercomparison. The identification of these sonde types within IGRA is based on the metadata collected by IGRA (i.e. the radiosonde code related to the WMO 3685 table) and on the use of the TAC code made available in the high-resolution BUFR files provided by an increasing number of stations since 2016 (about 1-2 hundred at present) and supplied for RHARM by ECWMF; these BUFR files include extensive metadata and a larger number of pressure levels for each reported radiosounding launch.

The selected radiosondes types are generally available since 2004 depending on the station. Each of these radiosondes has been post-processed using a combination of physical and statistical adjustments to remove various effects, e.g. the solar radiation bias or the bias due to the factory calibration. The post-processing applied to the radiosonde profiles allows the RHARM dataset to be a high-quality dataset which may be potentially used as a benchmark for intercomparisons and

validation purposes, including the validation of time series homogenization algorithms. The goal of the RHARM post-processed data discussed in the manuscript is to bridge the gap to a referencequality data processing like that provided by the GCOS Reference Upper-Air Network (GRUAN). We have improved the text wording in a few sections to further clarify these conceptual aspects.

The homogenization of earlier data is mentioned in this paper to create a link with a second paper, currently under preparation, describing the RHARM approach used for the radiosounding data before 2004, which is different and based on the use of a statistical method only and not with the help of the documented metadata. This is clarified in the introduction at the lines 127-129:

"b. Identification of change-points in the time series and adjustment of non-climatic (systematic) effects using statistical methods with related quantification of uncertainties in the historical observations.".

At very few stations only, where a full reconstruction of the utilized radiosonde types was possible, metadata are used for validating the identification of change-points in the time series. As a consequence, to have a complete metadataset at each station is not a mandatory requirement to process historical data using the RHARM approach, but surely provides value and increases the volume of information available for the validation of the results. Nevertheless, we clarify that all the selected RHARM stations have complete metadata since 2000 (i.e. stations reporting a radiosonde code, see also the WMO table 3685).

Secondly, the data set derived from this study is mostly only since 2014 and lacks lots of stations in USA, Russia and China (Fig. 1) so as to prevent its applications in the future.

First of all, it is important to clarify that there was a mistake in the maps of Figure 1 and 2, which have been now modified as follows.